# The enhancement of CCL2 and CCL5 by human bone marrow-derived mesenchymal stem/stromal cells might contribute to inflammatory suppression and axonal extension after spinal cord injury

Kazumichi Yagura[1,2], Hirokazu Ohtaki[1]*, Tomomi Tsumuraya[2], Atsushi Sato[2], Kazuyuki Miyamoto[3], Naoto Kawada[1], Keisuke Suzuki[1,3], Motoyasu Nakamura[1,3], Koji Kanzaki[2], Kenji Dohi[3], Masahiko Izumizaki[4], Yutaka Hiraizumi[5], Kazuho Honda[1]

1 Department of Anatomy, Showa University School of Medicine, Shinagawa-ku, Tokyo, Japan,
2 Department of Orthopedic Surgery, Showa University Fujigaoka Hospital, Fujigaoka, Aoba-ku, Yokohama, Kanagawa, Japan, 3 Department of Emergency & Clinical Care Medicine, Showa University School of Medicine, Shinagawa-ku, Tokyo, Japan, 4 Department of Physiology, Showa University School of Medicine, Shinagawa-ku, Tokyo, Japan, 5 Department of Orthopedic Surgery, Showa University School of Medicine, Shinagawa-ku, Tokyo, Japan

* taki@med.showa-u.ac.jp

## Abstract

Human bone marrow-derived mesenchymal stem/stromal cells (hMSCs) have shown potential in facilitating recovery from spinal cord injury (SCI) through communicating with microglia/macrophages (MG/MΦ). We here focused on chemokines as a candidate for the communication. Selected MG/MΦ-related chemokines were determined gene expression after SCI and further focused CCL2/CCR2 and CCL5/CCR5 to estimate role of the chemokines by hMSCs. Male C57/BL6 mice were subjected to spinal cord transection. Gene expression was assayed in the spinal cords following SCI for selected MG/MΦ-related chemokines and their receptors. hMSCs ($5 \times 10^5$ cells) were then transplanted into parenchyma of the spinal cord, and the expressions of the *Ccl2/Ccr2* and *Ccl5/Ccr5* axes, inflammation, MG/MΦ-polarization, and axonal regeneration were evaluated to measure the influence of the hMSCs. Finally, mouse CCL5 was injected into the spinal cords. Acute increases in gene expression after SCI were observed for most chemokines, including *Ccl2*; chronic increases were observed for *Ccl5*. CCL2+-cells merged with NeuN+-neurons. CCR2+ immunoreactivity was principally observed in Ly-6G+/iNOS+-granulocytes on postoperative day (pod) 1, and CCL5+ and CCR5+ immunoreactivity overlapped with NeuN+-neurons and F4/80+-MG/MΦ on pod 14. The hMSC transplantation enhanced *Ccl2* and *Ccl5* and improved locomotor activity. The hMSC implantation did not alter the number of Ly-6G+/CCR2+ but decreased *Il1*, *Elane*, and *Mpo* on pod 3. Conversely, hMSC transplantation increased expression of *Zc3h12a* (encodes MCP-1-induced protein) on pod 14. Moreover, hMSC increased the *Aif1*, and two alternatively activated macrophage (AAM)-related genes, *Arg1* and *Chil3* (Ym1), as well as axonal regenerative markers, *Dpysl2* and *Gap43*. Gene expression indicative of AAM polarization and axonal regeneration were partially recovered by

**Data Availability Statement:** All relevant data are within the paper and its Supporting Information files.

**Funding:** YES. This work was supported by KAKENHI, as follows: 16H05443 (Hirokazu Ohtaki), 17K10943 (Yutaka Hiraizumi), and 26861214 (Atsushi Sato).

**Competing interests:** The authors declare that they have no competing interests.

CCL5 injection. These results suggest that hMSC implantation increases *Ccl2* and *Ccl5*, improves locomotor activity, enhances MG/MΦ polarization to AAM, and increases the gene expression of axonal regenerative markers. These functions of hMSCs might be partially mediated by the CCL2/CCR2 and CCL5/CCR5 axes.

## Introduction

Every year, more than 10,000 people in the United States experience spinal cord injuries (SCI) due to trauma-related accidents. While medication regimens during the acute period of injury involve the extensive administration of steroids and other anti-inflammatory drugs, the recovery of neurologic function relies on the neural plasticity to each patient. The considerable number of patients that become permanently paralyzed despite medical intervention reflects the inadequacy of inherent mechanisms in repairing the nervous system [1, 2].

Bone marrow-derived mesenchymal stem/stromal cells (MSCs) are multipotent tissue stem cells that have shown promise in facilitating regeneration and/or recovery after neuronal injury [3–6]. We have previously demonstrated that xenografting human MSCs (hMSCs) into immunocompetent mice subjected to SCI [7] or brain ischemia [8] protected against short-term neurological deficits and damage; however, the levels of the grafted hMSCs decreased significantly within the first week. Further experiments indicated that hMSCs might communicate with recipient tissues and thereby convert the phenotypic polarization of host microglia/macrophages (MG/MΦ) to an alternatively activated phenotype (AAM), which is known to promote tissue repair and regeneration after injury [9]. Several immune- and nonimmune-related disease models and isogenic graft experiments have provided evidence for the immunomodulatory functions of hMSCs [10, 11]. However, precisely how hMSCs communicate with or regulate MG/MΦ remains unclear.

Chemokines guide the migration of immune cells, such as leukocytes and macrophages, and stimulate inflammatory responses, and are divided into the CC, CXC, C, and CX3C families from the structural feature [12]. Levels of many chemokines are increased in patients with central nervous system (CNS) disorders, such as SCI [13] and cerebral infarction [14], increase the levels of many chemokines. Indeed, enhanced levels of chemokines were observed in isogenic grafted bone marrow-derived MSCs of a hypoxia-ischemia rat model, but not in adipose tissue-derived MSCs [15]. Cultured hMSCs secrete several chemokines and express several chemokine receptors on the cell surface [16]. An additional study further reported that MSCs elevate the secretion of several chemokines by stimulating lipopolysaccharide (LPS) *in vitro* [17]. Hence, the communication between hMSCs and MG/MΦ may be partially mediated by chemokines.

Research has indicated that some chemokines play other roles in addition to chemotaxis response. Chemokines such as CCL2 (monocyte chemoattractant protein-1; MCP-1), CX3CL1 (fractalkine), CXCL12 (Stromal cell-derived factor 1; SDF-1), and CCL5 (regulated on activation, normal T cell expressed and secreted; RANTES) are reportedly involved in neuroprotection [18–21], axonal regeneration [22, 23], and the differentiation of MG/MΦ to AAM [24].

In present study, we firstly examined the effect and fate of transplanted hMSCs after SCI. Following the study, we determined gene expression for selected some MG/MΦ-related chemokines and their receptors on SCI, and then focused on two representative major MG/MΦ-related chemokine axes, CCL2/CCR2 and CCL5/CCR5, whose corresponding increases in the gene expression had been characterized as acute or chronic. The influences of hMSC

implantation following SCI on the CCL2/CCR2 and CCL5/CCR5 axes, inflammation, MG/
MΦ polarization, and/or axonal regeneration were examined to gain insight into the actions of
hMSCs.

## Materials and methods

### Animals

Wild-type, male C57BL/6 mice (8 to 12 weeks old, weighing 17 to 25 g) were purchased from
Sankyo Lab Service Corporation (Tokyo, Japan) and used for the experiments (total of 139 ani-
mals including 7 euthanized animals). Mice were maintained on a 12-h light/dark cycle at
$24 \pm 2°$ C with constant humidity ($55 \pm 10\%$) and free access to food and water. All experimen-
tal procedures involving animals were approved and followed by and conducted in accordance
with the guidelines of the Institutional Animal Care and Use Committee of Showa University
(#06105 and 07079).

### SCI model

The SCI mouse model was generated in accordance with previously described methods [7, 25].
Anesthesia was induced via inhalation of 4.0% sevoflurane and maintained with 3.0% sevoflur-
ane in $N_2O/O_2$ (2:1). Under aseptic conditions, an incision was made along the midline of the
skin on the back, and the muscles, soft tissues, and yellow ligaments overlying the spinal col-
umn between T9 and T10 were removed. The intervertebral spinal cord between T9 and T10
was then transected with a thin-bladed razor (FEATHER, Osaka, Japan). After bleeding had
stopped and the coagulated blood had been removed, the incision was closed, and the animals
were administered 1.0 mL of lactated Ringer's solution (s.c.; Otsuka, Tokyo, Japan) to prevent
dehydration of which were given once a day until they gain the body weight. To support their
eating, animal food was placed on the cage floor, and the water bottle was lowered to allow
easy access. Moreover, to support urination, the region of the lower abdomen in all mice was
gently stimulated a few times every day. Although we did not administer any pain relievers or
anesthesia during the experiments because they may influence SCI and locomotor activity, we
made all efforts to minimize animal numbers and animal discomfort. However, total of 7 ani-
mals were excluded from the study by humane endpoints. The 3-animals scored the Basso
Mouse Scale (BMS) [26] more than 3 immediately after SCI were excluded from the experi-
ments for incomplete hindlimb paralysis by a surgical error. Moreover, the 4-mice were
excluded from the study because the mice exhibited peritoneal infection, severe body weight
loss more than 25% before surgery, hindlimb wounds, and/or tail or foot autophagia. These
animals were euthanized by cervical dislocation or $CO_2$-gas exposure.

### Preparation of implanted hMSCs

Frozen vials of hMSCs from the bone marrow were obtained from Dr. Prockop (The Center
for the Preparation and Distribution of Adult Stem Cells [http://medicine.tamhsc.edu/irm/
msc-distribution.html]) under the auspices of a National Institutes of Health (NIH)/National
Center for Research Resources grant (P40 RR 17 447–06). The experiments were performed
with hMSCs from donor 281L [7, 8, 27]. To expand hMSCs, a frozen vial of $1.0\times10^6$ passage 3
cells was thawed and plated at 100 cells/cm$^2$ in multiple 150-mm plates (Nunclon, Thermo
Fisher Scientific, Waltham, MA) with a 20-mL complete culture medium (CCM) that con-
sisted of α-minimal essential medium (α-MEM; Invitrogen, Carlsbad, CA), 20% heat-inacti-
vated fetal bovine serum (FBS; Hyclone; Thermo Fisher Scientific), 100 units/mL penicillin,
100 μg/mL streptomycin (Invitrogen), and 2 mM L-glutamine (Invitrogen). The cultures were

incubated, and the medium was replaced every 3 days for approximately 8 days until the cells were 70–80% confluent. The medium was then discarded, and the culture plates were washed with phosphate-buffered saline (PBS). Adherent cells were harvested using 0.25% trypsin and 1 mM EDTA (Invitrogen) for 5 min at 37˚C and resuspended at $5 \times 10^5$ cells in 0.5 μL of sterile Hank's balanced salt solution (HBSS; Invitrogen) for injection.

### Injection of hMSCs into the spinal cord

The hMSCs were injected into the spinal cord in accordance with previously described methods [7]. Briefly, on the day following surgery to induce SCI, mice were re-anesthetized via inhalation of sevoflurane. The animals were placed face-down, and a truncated 29G needle with a 5.0-μL glass syringe (Hamilton, Reno, NV) was inserted directly into the intervertebral spinal cord (D/V -1.0 mm) between T10 and T11. We trained investigators to insert the needle to the depth of the anterior horn of the spinal cord by the injection of dye. Then, the depth of needle insertion was standardized at -1.0 mm from surface of which did not leak the dye from the needle tract during infusion or from the ventral spinal cord. The hMSCs ($5 \times 10^5$ cells/μL, n = 40 mice) or HBSS (n = 26 mice) was infused at a rate of 0.5 μL/min with an Ultra Micro Pump (World Precision Instruments, Sarasota, FL). Following infusion, the needle was left in place for 1 min to facilitate the diffusion of the solution into the tissue. Previously, we demonstrated that the fate of hMSCs does not differ between immunocompetent and immunodeficient animals after ischemia [8]. Therefore, in the present study, we refrained from using immunosuppressant following cell implantation [10].

### Assessment of motor function

Motor function after SCI was evaluated using an open-field behavior test that focused on hindlimb locomotor function according to the BMS. The BMS consists of an open-field locomotor rating scale that ranges from 0 (complete paralysis) to 9 (normal mobility). Briefly, individual mice were placed in the center of an open field ($50 \times 50$ cm$^2$) with a smooth, non-slip floor and monitored for 4 min. Hindlimb movements, trunk/tail stability, and forelimb-hindlimb coordination were assessed and graded. Mice were tested daily for 14 days after the operation. Mice were randomly numbered to ensure that the investigators remained unaware of the treatment groups during scoring.

### Isolation of RNA and production of cDNA

Under sodium-pentobarbital (100 mg/kg, i.p.) anesthesia, spinal cord samples from the T7 to T12 vertebrae were dissected on postoperative days (pod) 0 (sham-operated non-SCI control), 1, 3, 7, and 14 with/without hMSC or vehicle injection. The tissue was snap-frozen in liquid nitrogen and stored at −80˚C until use. Total RNA was isolated using the TRIZOL Reagent (Invitrogen) in accordance with the manufacturer's instructions and dissolved in RNase-free water. The purity and concentration of the extracted RNA were determined spectrophotometrically (NanoDrop, Wilmington, DE). A High-Capacity RNA-to-cDNA Kit (Applied Biosystems, Foster City, CA) was used to synthesize cDNA using 2 μg of total RNA in accordance with the manufacturer's instructions. To confirm the species specificity of the primers, we also extracted total RNA from cultured hMSCs and synthesized the cDNA.

### Semi-quantification of the number of hMSCs in the spinal cord

On days 1, 3, or 14 after SCI (0, 2, or 13 days after cell injection; n = 4 mice at each timepoint), mice were anesthetized with sodium pentobarbital (100 mg/kg, i.p.), and the spinal

cord was collected and subjected to cDNA preparation as described above. Semi-quantification of the number of hMSCs was determined based on the gene expression level of human *GAPDH* in mouse *Gapdh* via qPCR. Standard curves were generated by adding serial dilutions of hMSCs ($1 \times 10^2$ to $1 \times 10^6$ cells) to spinal cord samples (T7 to T12 vertebrae) of uninjured mice prior to homogenization according to the methods published by previous studies [7, 8, 28–30]. Finally, the estimated number of resident hMSCs was calculated from the standard curve. The human-specific primers cross-hybridized less than 0.07% with mouse cDNA obtained from spinal cord samples that had not been injected with hMSCs.

## Polymerase chain reaction (PCR)

PCR analyses were performed using TaKaRa Ex Taq (TaKaRa, Shiga, Japan) to confirm the species specificity of primers. The reaction mixture was created with an appropriate volume of the cDNA mixture: 0.25 µL of forward and reverse primers (50 nmol/mL), 2.0 µL of dNTP mixture (0.25 mM each), 0.1 µL of TaKaRa Ex Taq (5 units/µL), and 2.0 µL of $10 \times$ Ex Taq Buffer in a total volume of 20 µL. Thermal-cycling parameters were set as follows: 95˚ C for 1 min for initial denaturation, followed by a cycling regime of 40 cycles at 95˚ C for 45 s, 60˚ C for 30 s, and 72˚ C for 45 s. At the end of the final cycle, an additional 7-min extension step was conducted at 72˚ C. Ten microliters of each reaction mixture was electrophoresed onto 2.0% agarose gel, and the bands were visualized using ethidium bromide.

Quantitative PCR (qPCR) analyses were performed with SYBR Premix Ex Taq II reagent (TaKaRa) and an Applied Biosystems 7900HT Fast Real-Time PCR System (Applied Biosystems, Lincoln, CA). Relative gene expression was calculated using the absolute quantification method. The mouse glyceraldehyde-3-phosphate dehydrogenase gene (*Gapdh*) was used as a housekeeping gene to normalize the cDNA levels. All primers were designed as described in Table 1. All data are expressed the relative level (fold) to compare with pod 0 (basal level) after normalization against *mouse Gapdh* as a housekeeping gene (n = 6 mouse at each time-point).

## Immunostaining

Under sodium-pentobarbital (100 mg/kg, i.p.) anesthesia, the animals were transcardially perfused with 0.9% saline followed by 2% paraformaldehyde (PFA) in 50 mM phosphate buffer (PB) (pH 7.2), and the spinal cord (T7 to T12 vertebrae) was carefully removed. The spinal cords were then post-fixed overnight, immersed in 20% sucrose in 0.1 M PB for two nights, and embedded in liquid nitrogen-cooled isopentane using an embedding solution (20% sucrose in 0.1 M PB:O.C.T. compound [Sakura Finetech, Tokyo, Japan]: 2:1]. Sagittal spinal cord sections (thickness, 5 µm) were obtained using a cryostat (Hyrax50, Carl Zeiss, Inc.; Oberkochen, Germany) and stored at –80˚ C until use.

For single staining, the sections were washed with PBS and incubated in 0.3% $H_2O_2$ to quench internal peroxidase reactions. The sections were then immersed in 2.5% normal horse serum (NHS; Vector, Burlingame, CA) in PBS containing 0.05% Tween 20 (PBST) to block non-specific reactions for 1 h. Subsequently, the sections were incubated overnight with polyclonal rabbit anti-GFAP (1:10; DAKO, Glostrup, Denmark) or polyclonal rabbit anti-Iba1 antibodies (1:500; WAKO, Tokyo, Japan). After washing with PBST, the sections were incubated with biotinylated goat anti-rabbit immunoglobulin G (IgG) (1:200, Santa Cruz, Santa Cruz, CA) for 2 h. They were then incubated in an avidin-biotin complex solution (Vector) followed by diaminobenzidine (DAB; Sigma, St. Louis, MO) as a chromogen. Immunopositive cells were observed via light microscopy (AX70) using DP2-BSW image analysis software (Olympus, Tokyo, Japan).

**Table 1. Primers used for real-time polymerase chain reaction.**

| Name | Symbol | Species | Forward (5′ to 3′) | Reverse (5′ to 3′) | Size (bp) |
|------|--------|---------|--------------------|--------------------|-----------|
| AIF1 | *Aif1* | mouse | atcaacaagcaattcctcgatga | cagcattcgcttcaaggacata | 144 |
| Arg1 | *Arg1* | mouse | ctccaagccaaagtccttagag | aggagctgtcattagggacatc | 185 |
| CCL2 | *Ccl2* | mouse | ttaaaaacctggatcggaaccaa | gcattagcttcagatttacgggt | 121 |
| CCL3 | *Ccl3* | mouse | ttctctgtaccatgacactctgc | cgtggaatcttccggctgtag | 100 |
| CCL4 | *Ccl4* | mouse | ttcctgctgtttctcttacacct | ctgtctgcctcttttggtcag | 121 |
| CCL5 | *Ccl5* | mouse | gctccaatcttgcagtcgtg | gagcagctgagatgcccatt | 242 |
| CCL7 | *Ccl7* | mouse | gctgctttcagcatccaagtg | ccaggacaccgactactg | 135 |
| CCR1 | *Ccr1* | mouse | tggtgggcaatgtcctagtg | aagtaacagttcgggccctc | 300 |
| CCR2 | *Ccr2* | mouse | atccacggcatactatcaacatc | caaggctcaccatcatcgtag | 104 |
| CCR3 | *Ccr3* | mouse | accccgtacaacctggttct | ccaacaaaggcgtagattactgg | 158 |
| CCR5 | *Ccr5* | mouse | ttttcaagggtcagttccgac | ggaagaccatcatgttacccac | 158 |
| CRMP2 | *Dpysl2* | mouse | tcaaaggtggcaagattgtgaa | ggaatcaccattctggagtgg | 145 |
| CXCL1 | *Cxcl1* | mouse | ctgggattcacctcaagaacatc | cagggtcaaggcaagcctc | 117 |
| CXCL2 | *Cxcl2* | mouse | tccagagcttgagtgtgacg | tggttcttccgttgagggac | 201 |
| CXCR2 | *Cxcr2* | mouse | gggtggggagttcgtgtaga | aggtgctaggatttgagcct | 200 |
| ELANE | *Elane* | mouse | caggaacttcgtcatgtcagc | agcagttgtgatgggtcaaag | 162 |
| ENO2 | *Eno2* | mouse | tgagaataaatccttggagctggt | ggtcatcgcccactatctgg | 302 |
| GAP43 | *Gap43* | mouse | tctgctactaccgatgcagc | tggaggacggggagttatcag | 181 |
| GAPDH | *Gapdh* | mouse | gctacactgaggaccaggttgt | ctcctgttattatgggggtctg | 306 |
| GAPDH | *Gapdh* | human | ggagcgagatccctccaaaat | ggctgttgtcatacttctcatgg | 197 |
| GAPDH* | *Gapdh* | mouse | aggtcggtgtgaacggatttg | tgtagaccatgtagttgaggtca | 123 |
| IL-1β | *Il1b* | mouse | gcaactgttcctgaactcaact | atcttttggggtccgtcaact | 89 |
| IL-4 | *Il4* | mouse | tctcgaatgtaccaggagccatatc | agcaccttggaagccctacaga | 183 |
| MCPIP | *Zc3h12a* | mouse | acgaagcctgtccaagaatcc | taggggcctctttagccaca | 115 |
| MPO | *Mpo* | mouse | agggccgctgattatctacat | ctcacgtcctgataggcaca | 152 |
| Ym-1 | *Chil3* | mouse | ggtggacacagaatggttcttc | ccaggagcgttagtgacagc | 128 |

AIF: allograft inflammatory factor, Arg: arginase, CCL: chemokine (C-C motif) ligand, CCR: chemokine (C-C motif) receptor, Chil: chitinase-like, CRMP: collapsin response mediator protein, CXCL: chemokine (C-X-C motif) ligand, CXCR: chemokine (C-X-C motif) receptor, ELANE: neutrophil elastase, Eno: neuron-specific enolase, GAP: growth associated protein, GAPDH: glyceraldehyde-3-phosphate dehydrogenase, GFAP: glial fibrillary acidic protein, IL: interleukin, MCPIP: monocyte chemotactic protein-1 induced protein, MPO: myeloperoxidase, * = not species specific.

For multiple staining, sections that had been immersed in 0.3% $H_2O_2$ were incubated with an M.O.M. Immunodetection Kit (Vector), followed by 2.5% NHS/PBST. The sections were then incubated overnight with primary antibodies. The sections were rinsed with PBST and immersed with appropriate fluorescently labeled secondary antibodies for 2 h. Subsequently, cell nuclei were stained with 4,6-diamidine-2-phenylindole dihydrochloride (DAPI, 1:10,000; Roche, Manheim, Germany) and incubated in 1.0 mM $CuSO_4$ in 50 mM ammonium acetate buffer (pH 5.0) to diminish autofluorescence [31]. Fluorescence was detected using an Axio Imager optical sectioning microscope with ApoTome (Carl Zeiss, Inc.). All primary and secondary antibodies used for immunostaining are listed in Tables 2 and 3. For control staining, the same steps were performed with the exception of incubation with each primary antibody.

## Cell counting

Five sagittal sections (thickness: 5 μm) were collected from each mouse: at the midline (3), which included the central canal adjacent to the core injury site; and bilaterally at 150 μm (2 and 4) and 300 μm (1 and 5) lateral to the midline. The numbers of immunopositive cells

**Table 2. Primary antibodies used for immunohistochemistry.**

| Primary antibodies (Clone #) | Host | Companies | Catalog # | Dilution |
|---|---|---|---|---|
| CCL2 | Rabbit | Bioss (Boston, MA) | bs-1101R | 1,500 |
| CCL5 (25H14L17) | Rabbit | Thermo Fisher Scientific (Waltham, MA) | 701030 | 100 |
| CCL5 | Goat | R&D systems (Minneapolis, MN) | AF478 | 300 |
| CCR2 | Goat | Novus Biologicals (Littleton, Co) | NB100-701 | 300 |
| CCR2 | Goat | GeneTex (Irvine, CA) | GTX45788 | 300 |
| CCR5 | Rabbit | Bioss (Boston, MA) | bs-2514R | 500 |
| NeuN (A60) | Mouse | Merc Millipore (Burlington, MA) | MAB377 | 1,000 |
| GFAP | Rabbit | DAKO (Glostrup, Denmark) | N1506 | 10 |
| GFAP (G-A-5) | Mouse | SIGMA (St. Louis, MO) | G3893 | 1,000 |
| Iba1 | Rabbit | Wako (Osaka, Japan) | 019–19741 | 500 |
| Alexa635labeled-Iba1 | Rabbit | Wako (Osaka, Japan) | 013–26471 | 500 |
| F4/80 (Cl:A3-1) | Rat | BMA Biomedicals (Augst, Switzerland) | T-2008 | 500 |
| Ly-6G (RB6-8C5) | Rat | BD Biosciences (Franklin Lakes, NJ) | 557445 | 400 |
| iNOS | Rabbit | Merc Millipore (Burlington, MA) | 06–573 | 250 |
| Ym-1 | Rabbit | StemCell Technologies (Vancouver, BC Canada) | 1404 | 200 |
| CRMP2 | Rabbit | LifeSpan BioSciences (Seattle, WA) | LS-B5657 | 3,000 |
| GAP43 (EP890Y) | Rabbit | Abcam (Cambridge, UK) | ab75810 | 500 |

CCL: chemokine (C-C motif) ligand, CCR: chemokine (C-C motif) receptor, CRMP: collapsin response mediator protein, GAP: growth associated protein, GFAP: glial fibrillary acidic protein, Iba1: ionized calcium-binding adapter molecule, iNOS: inducible nitric oxide synthase, NeuN: neuronal nuclei.

against CCR2, Iba1, Ly-6G, and inducible nitric oxide (iNOS) antibodies were counted in four random fields at the injury site in each animal at 500× magnification and expressed as the average total number in a square millimeter ($mm^2$).

## Injection of recombinant mouse CCL5 (RANTES) into the spinal cord

To investigate the role of CCL5 in the spinal cord after injury, recombinant mouse CCL5 was directly injected into the spinal cord between T10 and T11 according to the cell-injection procedure described above. Recombinant mouse CCL5 (RANTES; Miltenyi Biotec, Bergisch Gladbach, Germany) was dissolved in saline containing 0.1% bovine serum albumin (BSA; Wako) under aseptic conditions to obtain a concentration of 20 ng/μL. Seven days after SCI, mouse spinal cords were injected with a bolus of CCL5 (20 ng, n = 6 mice) or vehicle (n = 6 mice) at a rate of 0.5 μL/min. Mice were anesthetized with sodium pentobarbital (100 mg/kg, i. p.) at 24 h following injection, and spinal cord tissue was dissected from T7 to T12 for qPCR.

**Table 3. Secondary antibodies used for immunohistochemistry.**

| Secondary antibodies | Host | Companies | Catalog # | Dilution |
|---|---|---|---|---|
| Rabbit IgG (biotin) | Goat | Santa Cruz Biotechnology (Santa Cruz, CA) | SC-2040 | 200 |
| Goat IgG (biotin) | Horse | Vector Laboratory (Burlingame, CA) | BA-9500 | 200 |
| Mouse IgG (Alexa 546) | Goat | ThermoFisher Scientific (Waltham, MA) | A11030 | 800 |
| Rabbit IgG (Alexa 488) | Goat | ThermoFisher Scientific (Waltham, MA) | A11034 | 400 |
| Rabbit IgG (Alexa 546) | Goat | ThermoFisher Scientific (Waltham, MA) | A11035 | 400 |
| Rabbit IgG (Alexa 647) | Goat | ThermoFisher Scientific (Waltham, MA) | A21245 | 400 |
| Rat IgG (FITC) | Goat | Jackson ImmunoResearch (West Grove, PA) | 112-096-071 | 200 |
| Goat IgG (Alexa FITC) | Donkey | ThermoFisher Scientific (Waltham, MA) | A11055 | 400 |
| Goat IgG (Alexa 546) | Donkey | ThermoFisher Scientific (Waltham, MA) | A1056 | 400 |

## Statistical analysis

As each mouse was assigned a random number, all data were collected and analyzed without the investigator being aware of the group identities. Data are expressed as the mean ± standard error of the mean (SEM). Statistical comparisons were conducted with one-way analyses of variance (ANOVAs) following non-parametric multiple comparisons, as indicated in each figure legend. A value of $P < 0.05$ was considered to indicate statistical significance. Analyses were performed using Statcel4 (OMS Ltd., Tokyo, Japan).

## Results

### Injection of hMSCs suppressed SCI symptoms

We have previously reported that transplant viable hMSCs, but not inviable hMSCs into injured spinal cord significantly improved locomotor activity and decreased lesion size which is surrounded by GFAP immunoreactions within a week [7]. These effects were likely mediated through MG/MΦ-hMSC communication. To confirm the reproducibility of these findings, we determined the BMS score, the damaged site by performing GFAP immunostaining and assessing the gene expression of *Eno2* (neuron-specific enolase) as well as the retention of hMSCs in the spinal cord (Fig 1).

On day 1 after SCI, hMSCs ($5 \times 10^5$ cells/µL) or HBSSs were injected into the intervertebral spinal cord. To validate the injection and retention of the hMSCs, human *GAPDH* was semi-quantified via qPCR, and the estimated hMSC number was calculated from the standard curve (Fig 1A). Immediately after hMSC injection (pod 1), approximately 260,000 hMSCs were

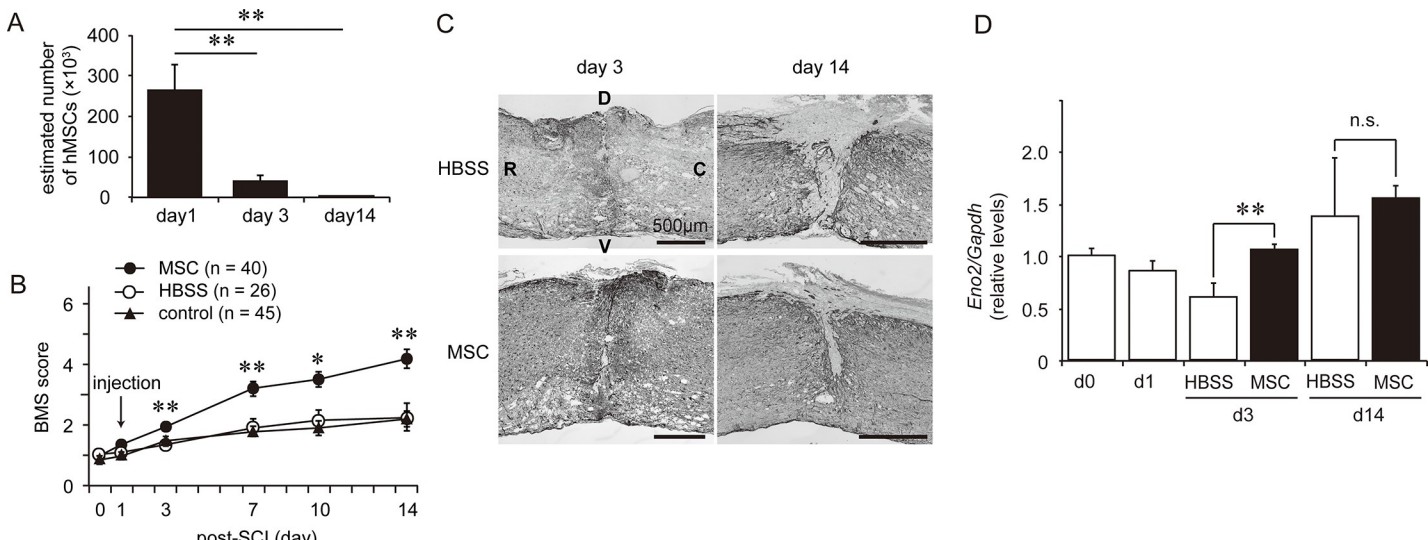

**Fig 1. Injection of human mesenchymal stem/stromal cells (hMSCs) suppresses spinal cord injury (SCI) symptoms. (A)** Retention of hMSCs after injection into the spinal cord 1 day after SCI (n = 4 mouse at each time-point). **: $P < 0.01$ (Dunnett's *post hoc* test, *vs* day 1). **(B)** Locomotor activity was determined based on BMS scores after SCI with/without hMSC injection. We injected hMSCs (filled circles, n = 40 mouse) or vehicle (HBSS; open circles, n = 26 mouse) into the spinal cord on postoperative day (pod) 1. Locomotor activity was then tracked for 14 days. Control animals (filled triangle, n = 45 mouse) received no injection into the spinal cord after SCI. *: $P < 0.05$, **: $P < 0.01$ (Tukey's *post hoc* test). **(C)** Representative images of lesion site in the spinal cord 3 and 14 days after SCI (n = 3 mouse in each group). The lesion site was identified as the area surrounded by enhanced GFAP expression. Vehicle-treated control animals (upper) exhibited a larger lesion area than hMSC-treated animals (bottom). R: rostral, C: caudal, D: dorsal, V: ventral. **(D)** Expression of *Eno2* was quantified via real time-PCR after SCI with/without hMSCs in a time-dependent fashion. *Eno2* expression was temporarily decreased on pod 3 in the HBSS-treated group (n = 5 mouse). Significant suppression of *Eno2* expression was observed in the hMSC-treated group (n = 10 mouse) on pod 3. **: $P < 0.01$ (Dunnett's *post hoc* test), n.s.: not significant. The gene expression levels are expressed the relative level (fold) to compare with pod 0 (basal level, n = 6 mouse) after normalization against *Gapdh* as a housekeeping gene. All data are expressed as the mean ± SEM.

detected–representing 50% of the injected cells–at the five vertebral sites (Th 7–12), including the site of injection. The numbers of hMSCs decreased to approximately 40,000 cells on pod 3, and have few-to-no hMSCs were detected on pod 14.

SCI induced severe locomotor disability, with BMS scores reaching approximately 1.5 points within a few hours; they gradually increased through pod 14. Animals treated with hMSCs were associated with significantly higher BMS scores than were the vehicle-treated animals from pod 3 through pod 14 (Fig 1B). Typical images of the lesion sites surrounded by GFAP immunoreactions in the hMSC-treated and HBSS-treated groups on pod 3 and 14 are shown in Fig 1C. The hMSC-treated mice exhibited fewer lesions (i.e., glial scarring) than the HBSS-treated mice at pod 14.

Eno2 is a neuron-specific glycolytic enzyme which is considered to indicate neuronal viability. Expression of *Eno2* was significantly higher in hMSC-treated animals than in HBSS-treated animals (Fig 1D). All of these results support the findings of our previous study [7]: hMSCs suppress SCI-induced adverse events.

## Expression of chemokines and their receptors after SCI

We next quantified the levels of MG/MΦ-related chemokines and the expression of their receptor genes in mice during the experimental periods without hMSC injection after the SCI (Fig 2). The expression of selected chemokine genes, including *Ccl2*, *Ccl3*, *Ccl4*, *Ccl7*, *Cxcl1*, and *Cxcl2*, significantly increased by 48-, 41-, 228-, 52-, 17-, and 56-fold, respectively, on pod 1. With the exceptions of *Ccl3*, *Ccl4*, and *Cxcl1*, decreases in gene expressions were mostly observed on pod 7. Levels of *Ccl3*, *Ccl4*, and *Cxcl1* expression remained 41-, 112-, and 7-fold

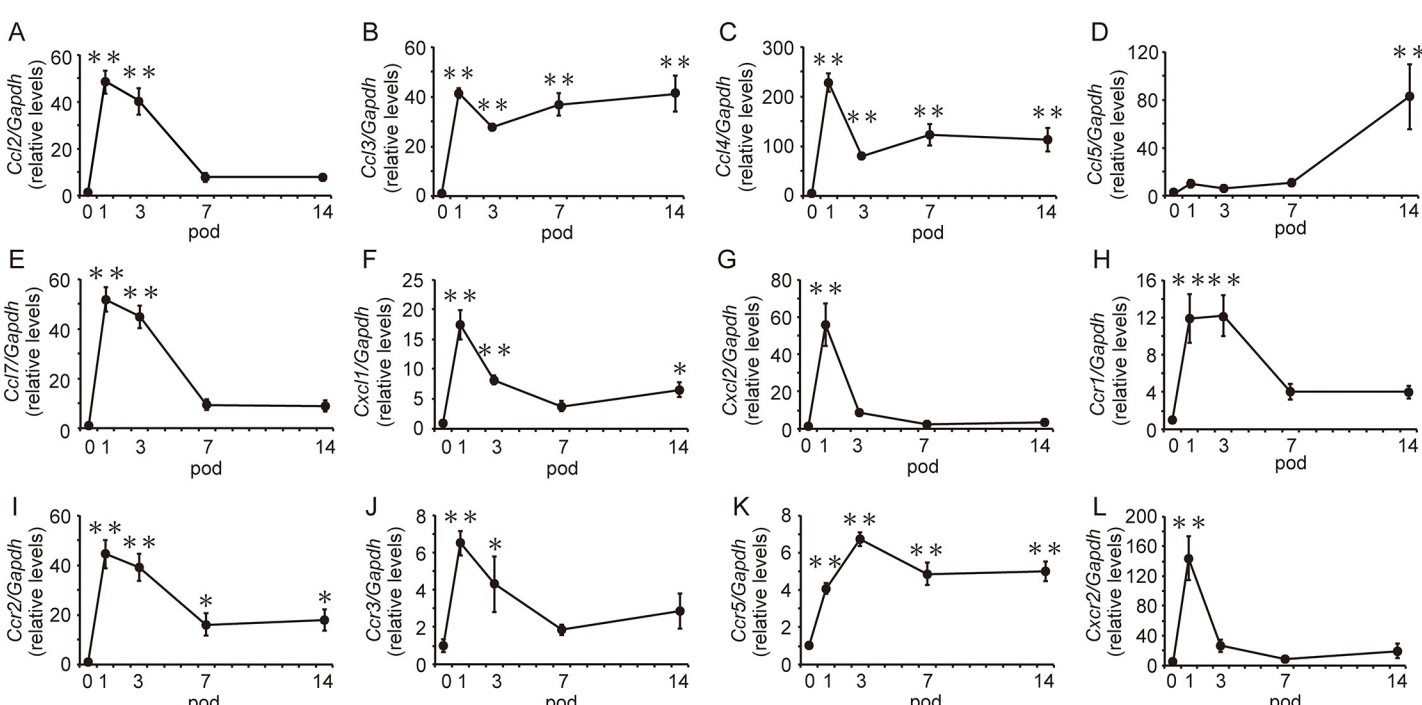

**Fig 2. Gene expression of chemokines and their receptors after spinal cord injury (SCI).** Expression of *Ccl2* **(A)**, *Ccl3* **(B)**, *Ccl4* **(C)**, *Ccl7* **(E)**, *Cxcl1* **(F)**, and *Cxcl2* **(G)** significantly increased on post-operative day (pod) 1. In contrast, increases in *Ccl5* **(D)** expression were observed on pod 14. Increases in gene expression were observed for all chemokine receptors **(H—L)** on pod 1. While *Ccr1* **(H)**, *Ccr3* **(J)**, and *Cxcr2* **(L)** expression decreased by pod 7, *Ccr2* **(I)** and *Ccr5* **(K)** levels remained significantly elevated on pod 14. All data are expressed the relative level (fold) to compare with pod 0 (basal level) after normalization against *Gapdh* as a housekeeping gene (n = 6 mouse at each time-point). Data are expressed as mean ± SEM. *: $P<0.05$, **: $P<0.01$ (Dunnett's *post hoc* test, *vs* day 0).

greater than pre-injury levels on pod 14. Although no increases in *Ccl5* expression were observed during the acute period, levels of *Ccl5* expression had increased by 82-fold on pod 14. By contrast, the expression of all chemokine receptors, including *Ccr1*, *Ccr2*, *Ccr3*, *Ccr5*, and *Cxcr2* significantly increased by 12-, 44-, 6-, 4-, and 144-fold, respectively, on pod 1. Although *Ccr1*, *Ccr3*, and *Cxcr2* expression decreased on pod 7, *Ccr2* and *Ccr5* levels remained 18-fold and 5-fold greater, respectively, at pod 14. We divided these chemokines into two groups: those exhibiting acute increases following SCI (CCL2, CCL3, CCL4, CCL7, CXCL1, and CXCL2) and those exhibiting increases in the chronic phase after SCI (CCL5). Then, we further analyzed the cellular localization of CCL2 and CCL5 and their receptors in the spinal cord via immunohistochemistry because they have been reported well for MG/MΦ migration as a representative gene.

## Distribution of CCL2 and CCR2 expression in the spinal cord on day 1

We examined the cellular localization of CCL2 and its receptor CCR2 via immunohistochemistry before inducing SCI and at pods 1 and 14 after SCI. Few-to-no CCL2 immunoreactions were observed in the spinal cord of pre-SCI mice. Immunopositive reactions for CCL2 were observed around the site of SCI at pod 1. CCL2 immunoreactions at pod 1 merged with NeuN (a neuronal marker) but not with GFAP (an astrocytic marker) or F4/80 (an MG/MΦ marker) (Fig 3A and 3B). Similar to the findings observed for *Ccl2* expression (Fig 2A), CCL2 immunoreactivity had diminished by pod 14 (Fig 3C).

Few-to-no CCR2 immunoreactions were observed in the spinal cord pre-SCI. However, immunoreactivity was obviously recognized at the epicenter of injury on pod 1 but diminished

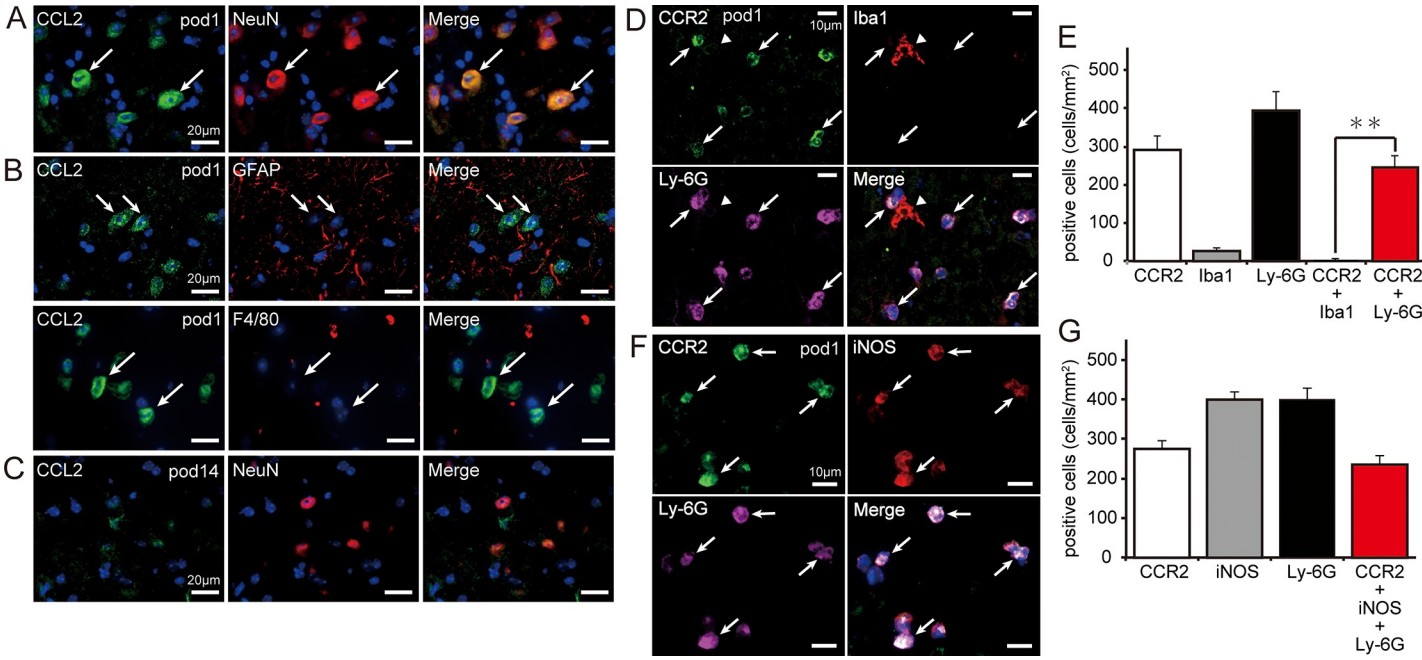

**Fig 3. Localization of CCL2 and CCR2 after spinal cord injury (SCI). (A)** CCL2+ cells (*green*) were observed around the site of SCI on day 1 and were merged with the neuronal marker NeuN (*red*) (arrows). **(B)** CCL2+ immunoreactions (*green* with arrows) did not overlap with the astrocyte marker GFAP (upp*e*r in *red*) or the MG/MΦ marker F4/80 (bottom in *red*). **(C)** The intensity of CCL2+ immunoreactivity decreased by postoperative day (pod) 14. **(D)** CCR2+ immunoreactivity (*green*) was mainly expressed in Ly-6G+ cells (*pink*) around the site of the spinal cord lesion (arrows), but not in Iba1+ cells (*red*) (arrowhead). **(E)** Numbers of CCR2+, Ly-6G+, Iba1+, and merged cells were counted at the site of injury (n = 5 mouse). Data are expressed as the mean ± SEM. **: *P<0.01* (Dunnett's *post hoc* test). **(F)** Many CCR2+ (*green*)/Ly-6G+ (*pink*) cells were also merged with iNOS (*red*) (arrows). **(G)** Numbers of CCR2+, Ly-6G+, iNOS+, and merged cells were counted (n = 5 mouse). Most CCR2+ cells were co-stained with Ly-6G and iNOS. Data are expressed as the mean ± SEM. *Blue* indicates nuclear staining with DAPI.

once again by pod 14. These results were consistent with those observed regarding *Ccr2* expression (Fig 2I). Most CCR2-positive (CCR2+) cells exhibited a round shape and a segmented nucleus. Multiple-staining experiments revealed that CCR2+ cells had merged, suggesting that these cells were granulocytes (Fig 3D). Few-to-no CCR2+ reactions were observed in cells stained with Iba1, NeuN, and GFAP. The numbers of CCR2+, Ly-6G+, and Iba1+ cells at the site of injury were $292 \pm 32$, $394 \pm 42$, and $27 \pm 6$ cells/mm$^2$, respectively. Moreover, we observed $245 \pm 26$ cells/mm$^2$ CCR2+/Ly-6G+ cells at the site of injury: a significantly greater number than that of CCR2+/Iba1+ cells ($4 \pm 2$ cells/mm$^2$) (Fig 3E). Our results also indicated that CCR2+/Ly-6G+ cells had merged with iNOS+ cells ($236 \pm 21$ cells/mm$^2$) (Fig 3F and 3G).

### Distribution of CCL5 and CCR5 expression in the spinal cord on day 14

We aimed to determine the localization of CCL5 and its receptor. Previous studies have reported that CCL5 binds to three receptors: CCR1, CCR3, and CCR5 [32]. Because *Ccr1* and *Ccr3* expression temporarily increased in the acute phase and decreased by pod 7, we focused only on the levels of CCR5, whose underlying gene expression exhibited sustained increases by pod 14 (Fig 2H, 2J and 2K). Few-to-no CCL5 immunoreactions were observed in the spinal cord immediately following SCI or pod 1. However, CCL5 immunoreactions were observed on pod 14 around the SCI site. The CCL5 immunoreactions merged with NeuN+ and F4/80+ cells but not with GFAP+ cells (Fig 4A).

Although the immunoreactions were prominent at the epicenter of the injury on pod 1, CCR5 immunoreactivity was rarely observed post-SCI. CCR5+ cells exhibited a round shape with lobed nuclei; moreover, similar to the findings observed for CCR2 on pod 1, CCR5+ cells were merged with Ly-6G (Fig 4B). However, unlike the findings for CCR2, CCR5

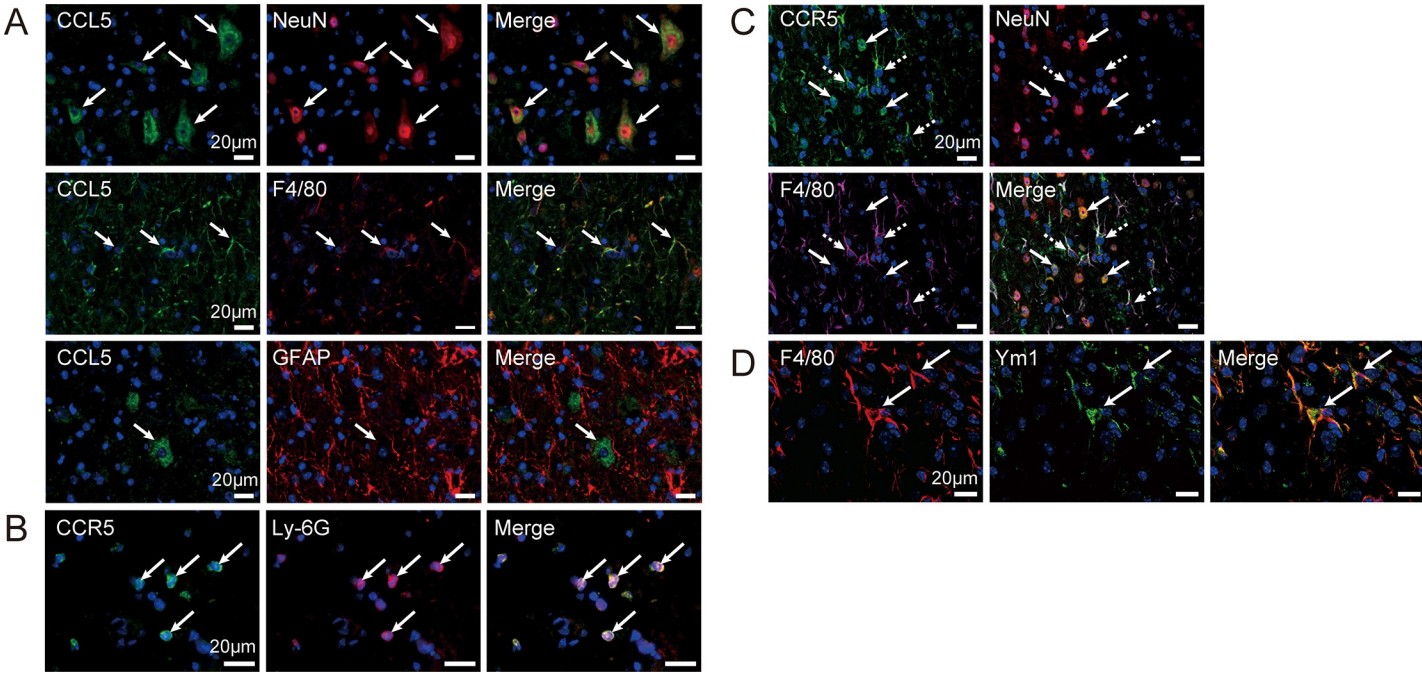

**Fig 4. Localization of CCL5 and CCR5 after spinal cord injury (SCI). (A)** CCL5+ cells were observed at the peri-injury site on pod 14 and were merged with NeuN+ (*red*) and F4/80+ (*red*) cells, respectively (arrows). However, CCL5+ cells were not merged with GAFP+ cells (*red*). **(B)** CCR5+ immunoreactivity (*green*) was observed in the epicenter on pod 1, and merged with Ly-6G+ cells (*red*) (arrows). **(C)** CCR5+ cells (*green*) were observed at the peri-injury site on pod 14, and were merged with NeuN+ (*red* with arrows) and F4/80+ (arrows with dashed lines) cells. **(D)** F4/80+ cells (*red*) on pod 14 co-stained with Ym1+ immunoreactions (*green*), a marker for M2-type alternatively activated MG/MΦ (arrows). Nuclei were detected using DAPI.

immunoreactivity was still observed on pod 14 at the peri-injury site. Furthermore, CCR5[+] cells were merged with NeuN+ and F4/80+ immunoreactions (Fig 4C) on pod 14. Moreover, the F4/80[+] cells were co-stained with Ym1, which is an AAM marker (Fig 4D).

## Effects of hMSCs on CCL2, CCR2, and the inflammatory response

We subsequently investigated the effect of hMSC treatment on the levels of gene expression corresponding to chemokines and their receptors. The mouse specificity of the chemokine primers was examined using mouse spinal cords and naïve hMSCs via RT-PCR (Fig 5A). A house-keeping gene, *Gapdh* (glyceraldehyde-3-phosphate dehydrogenase), primers for which amplified both human and mouse samples gene expression in our previous study [27], was recognized signals from both mouse and human samples. However, all tested primers for *Ccl2*, *Ccl5*, and their receptors were amplified with estimated size in mouse spinal cord samples (Fig 5A). We then compared *Ccl2* levels in the spinal cord between mice injected with HBSS or hMSCs after SCI. *Ccl2* expression on pod 3 was approximately 16-fold higher in mice treated with hMSCs than in those treated with HBSS; these increases remained significant at pod 14 (Fig 5B). Although levels of *Ccr2* expression in the spinal cord were similar in hMSC- and HBSS-treated mice on pod 3, they increased in hMSC-treated mice by pod 14 (Fig 5C).

No obvious differences were observed in CCL2[+] or CCR2[+] staining, which had merged with NeuN staining around the site of injury, between the groups on pod 3. However, the intensity of immunoreactivity tended to be greater in mice treated with hMSCs (Fig 5D). While Ly-6G[+] cells were easily recognized at the sites of injury, their numbers did not differ significantly between the groups. However, there were few CCR2[+] cells, and they did not overlap with Ly-6G[+] cells (Fig 5E and 5F). On pod 3, CCR2 immunoreactivity was observed in larger cells at the site of injury, overlapping with NeuN[+] cells (Fig 5G).

To determine the effect of hMSC treatment on inflammation involving granulocytes, we examined gene expression for inflammatory markers, such as *Il1b*, *Elane* (neutrophil expressed elastase), and *Mpo* (myeloperoxidase) on pod 3. While HBSS-injected SCI mice exhibited an increase in the expression of all three genes in the spinal cord, this expression was significantly suppressed in mice treated with hMSCs (Fig 5H–5J). *Zc3h112a* (Zinc finger CCCH-type containing 12A, also known as monocyte chemotactic protein-induced protein) is a protein whose expression is induced by CCL2; *Zc3h112a* is thereby implicated in immunosuppression and M2-type macrophage polarization [33]. Mice treated with hMSCs exhibited significant increases in *Zc3h112a* expression in the spinal cord on pod 14 (Fig 5K).

## Effects of hMSCs on CCL5 and CCR5 expression

We next investigated the effects of hMSCs on CCL5 and CCR5 at pod 14. We observed no significant differences in the expressions of *Ccl5* or *Ccr5* in the spinal cord between the hMSC- and HBSS-treated groups on pod 3. Although no differences in *Ccr5* expression were observed, *Ccl5* expression in the spinal cord increased by approximately 11-fold in hMSC-treated mice by pod 14 relative to the levels observed in HBSS-treated mice (Fig 6A and 6B).

Immunostaining experiments confirmed that CCL5 merged with NeuN[+] and F4/80[+] cells (Fig 3). However, the CCL5[+]/F4/80[+] cells were more dominant in hMSC-treated mice (Fig 6C). In hMSCs-treated mice, CCR5[+] cells were merged with F4/80[+] cells around the site of injury (Fig 6D).

## Influence of hMSCs on MG/MΦ polarization

Injection of hMSCs on pod 1 influenced MG/MΦ levels on pod 14. Quantification experiments revealed that hMSC-treated animals exhibited significantly greater *Aif1* (a pan-MG/MΦ

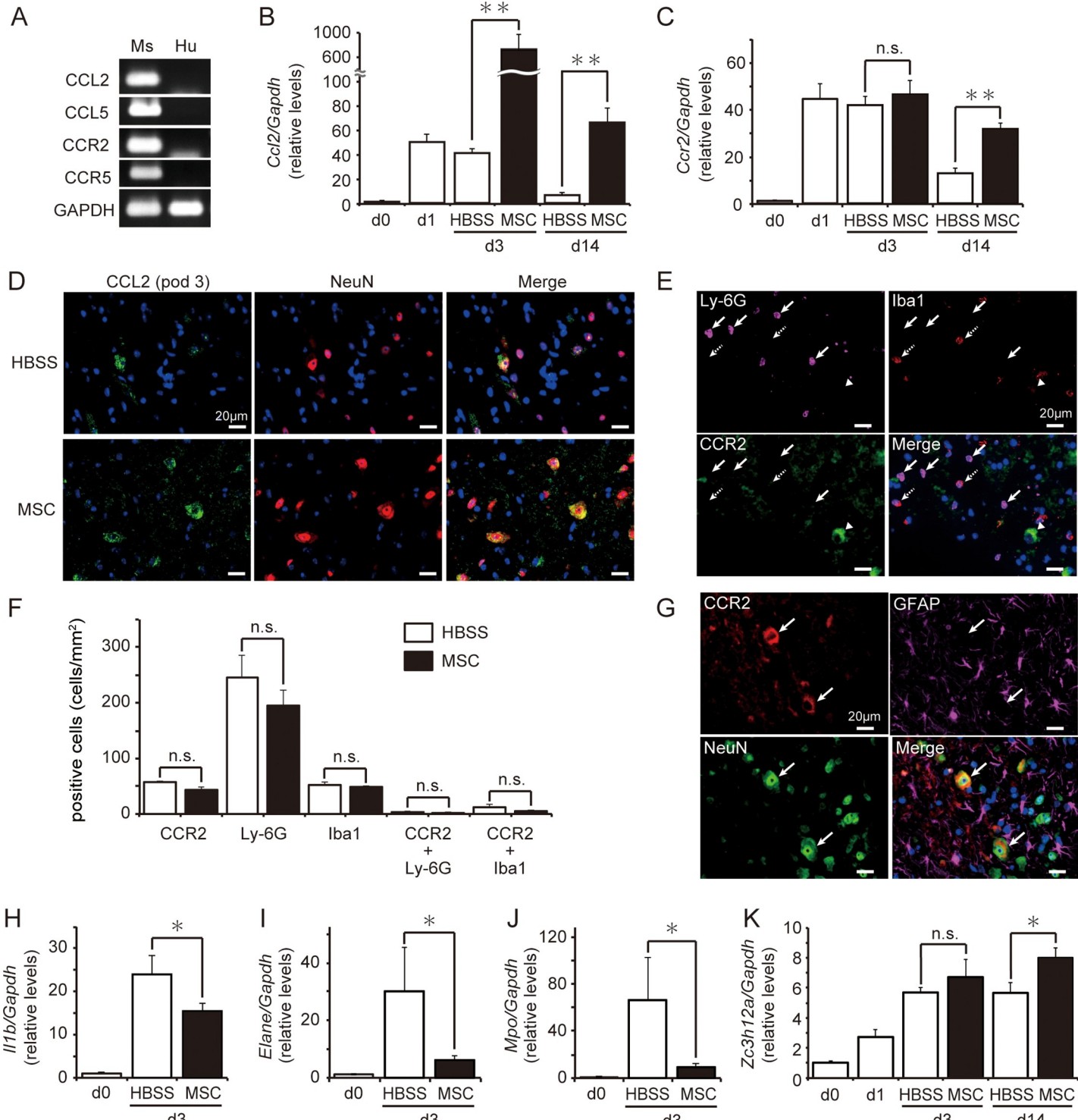

**Fig 5. Human mesenchymal stem/stromal cells (hMSCs) increased *Ccl2* expression and reduced inflammatory responses. (A)** All tested chemokine and receptor primers were detected at the expected rates in mouse samples (Ms). *Ccl2* **(B)** and *Ccr2* **(C)** expressions in the spinal cord were quantified via real time-PCR after spinal cord injury (SCI) and hMSC injection. *Ccl2* levels were significantly increased in hMSC-treated mice (n = 19 mouse) on postoperative days (pod) 3 and 14 relative to the levels observed in HBSS-treated mice (n = 13 mouse) on the same pod. Temporary increases in *Ccr2* expression were observed in HBSS-treated mice, with levels decreasing by pod 14. However, increases in *Ccr2* expression remained significant in hMSC-treated mice on pod 14. Data are expressed as mean ± SEM. **: $P < 0.01$ (Dunnett's *post hoc* test), n.s.: not significant. **(D)** CCL2+ cells (*green*) were co-labeled with NeuN+ cells (*red*) in both groups on pod 3. *Blue* indicates nuclear staining using DAPI. **(E)** Slight CCR2+ immunoreactivity (*green*) was observed in Ly-6G+ (*pink* with arrows) and Iba1+ (arrows with dashed lines) cells. Moreover, some large

cells were labeled with CCR2 antibody (arrowhead). *Blue* indicates nuclear staining using DAPI. **(F)** No significant differences in the number of positive cells were observed between the hMSCs- and HBSS-treated groups (n = 3 mouse at each group). **(G)** Large CCR2⁺ cells (*red*; arrows) co-localized with NeuN⁺ reactions (*green*) but not with GFAP⁺ reactions (*pink*). *Blue* indicates nuclear staining using DAPI. **(H-K)** The transplantation of hMSCs influenced inflammation-induced gene expression in the spinal cord on pod 3. Animals treated with hMSCs (n = 10 mouse) exhibited significant decreases in *Il1b* **(H)**, *Elane* **(I)**, and *Mpo* **(J)** expression in the spinal cord on pod 3 relative to HBSS-treated animals (n = 5 mouse). **(K)** Conversely, *Zc3h12a* expression was significantly greater in hMSC-treated mice (n = 9 mouse) than in HBSS-treated mice (n = 8 mouse) on pod 14. The gene expression levels are expressed the relative level (fold) to compare with pod 0 (basal level, n = 6 mouse) after normalization against *Gapdh* as a housekeeping gene. Data are expressed as mean ± SEM. *: $P < 0.05$ (Dunnett's *post hoc* test), n.s.: not significant.

marker, also known as Iba1) expression than did HBSS-treated animals on pod 14 (Fig 7A). Immunostaining for Iba1 in the spinal cord revealed that Iba1⁺ reactions clustered at both the peri-injury and injection sites in hMSC-treated mice; no such clustering was observed around the site of injection in the HBSS-treated group (Fig 7B). Fluorescent staining revealed that F4/80⁺ MG/MΦ-like cells were co-expressed with Ym1⁺ cells around the injection site, suggestive of M2-type AAMs (Fig 7C). Moreover, the number of F4/80⁺ MG/MΦ-like cells was greater in hMSC-treated mice than in HBSS-treated mice.

Expression of the AAM markers *Arg1* (*Arginase1*) and *Chil3*, which encodes Ym1, was significantly lower in the hMSCs-treated group than in the HBSS-treated group on pod 3. However, levels of gene expression had increased by 15- and 2-fold, respectively, in the hMSCs-treated group on pod 14 (Fig 7D and 7E). Moreover, the expression of the anti-inflammatory cytokine *IL-4* had increased by 3-fold in hMSCs-treated animals by pod 14 (Fig 7F).

## Influence of hMSCs on axonal regeneration

To determine the effect of hMSCs on axonal regeneration, we examined gene expression and protein localization corresponding to CRMP2 (encoded by *Dpysl2*) and GAP43 in the mouse spinal cord after SCI. Expression of *Dpysl2* remained nearly unchanged in HBSS-treated mice

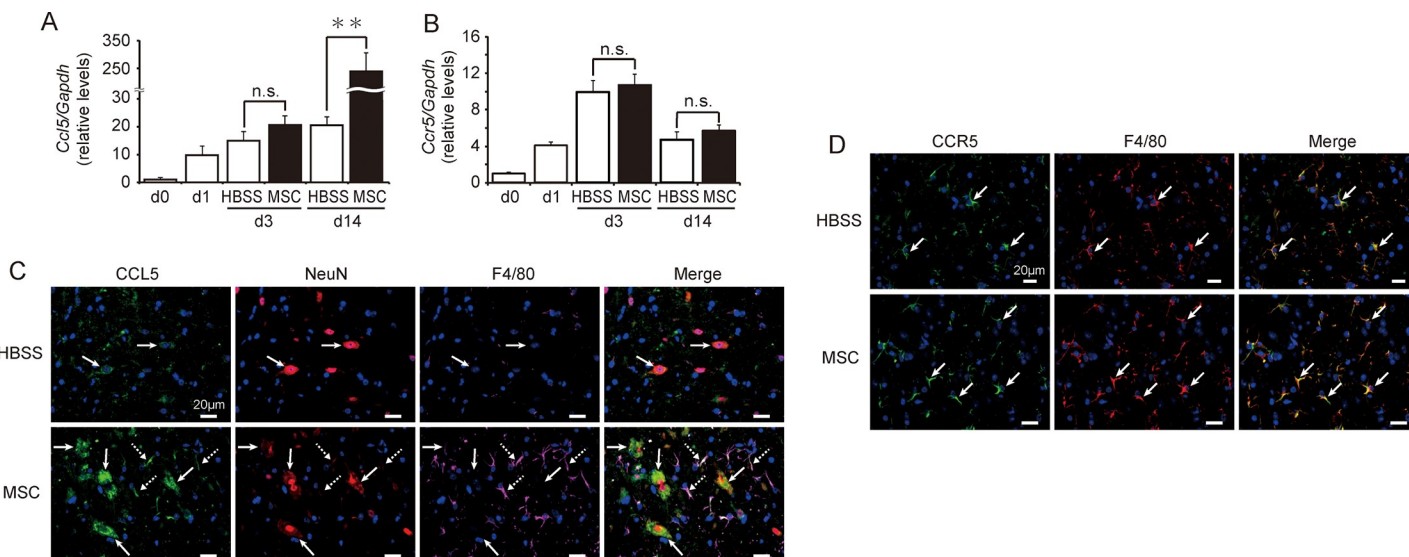

**Fig 6. Human mesenchymal stem/stromal cells (hMSCs) increased levels of CCL5 expression during the chronic phase.** Expression of mouse **(A)** *Ccl5* and **(B)** *Ccr5* in the spinal cord was quantified via real-time PCR in a time-dependent fashion after spinal cord injury (SCI) and hMSC injection. Mice treated with hMSCs (n = 19 mouse) exhibited significant increases in *Ccl5* expression relative to HBSS-treated mice (n = 13 mouse) on postoperative day (pod) 14. However, no significant differences in *Ccr5* expression were observed between the two groups during the experimental period. Data are expressed the relative level (fold) to compare with pod 0 (basal level, n = 6 mouse) after normalization against *Gapdh* as a housekeeping gene. Data are expressed as the mean ± SEM. **: $P < 0.01$ (Dunnett's *post hoc* test), n.s.: not significant. **(C)** CCL5 immunoreactivity (*green*) was observed at the peri-injury site on pod 14 and merged with NeuN⁺ (*red*) and F4/80⁺ (*pink*) cells. Mice treated with hMSCs exhibited greater CCL5⁺ immunoreactivity than did HBSS-treated mice. **(D)** Similar CCR5⁺ reactions (*green*) were merged with F4/80⁺ (*red*) cells in both experimental groups. *Blue* indicates nuclear staining using DAPI.

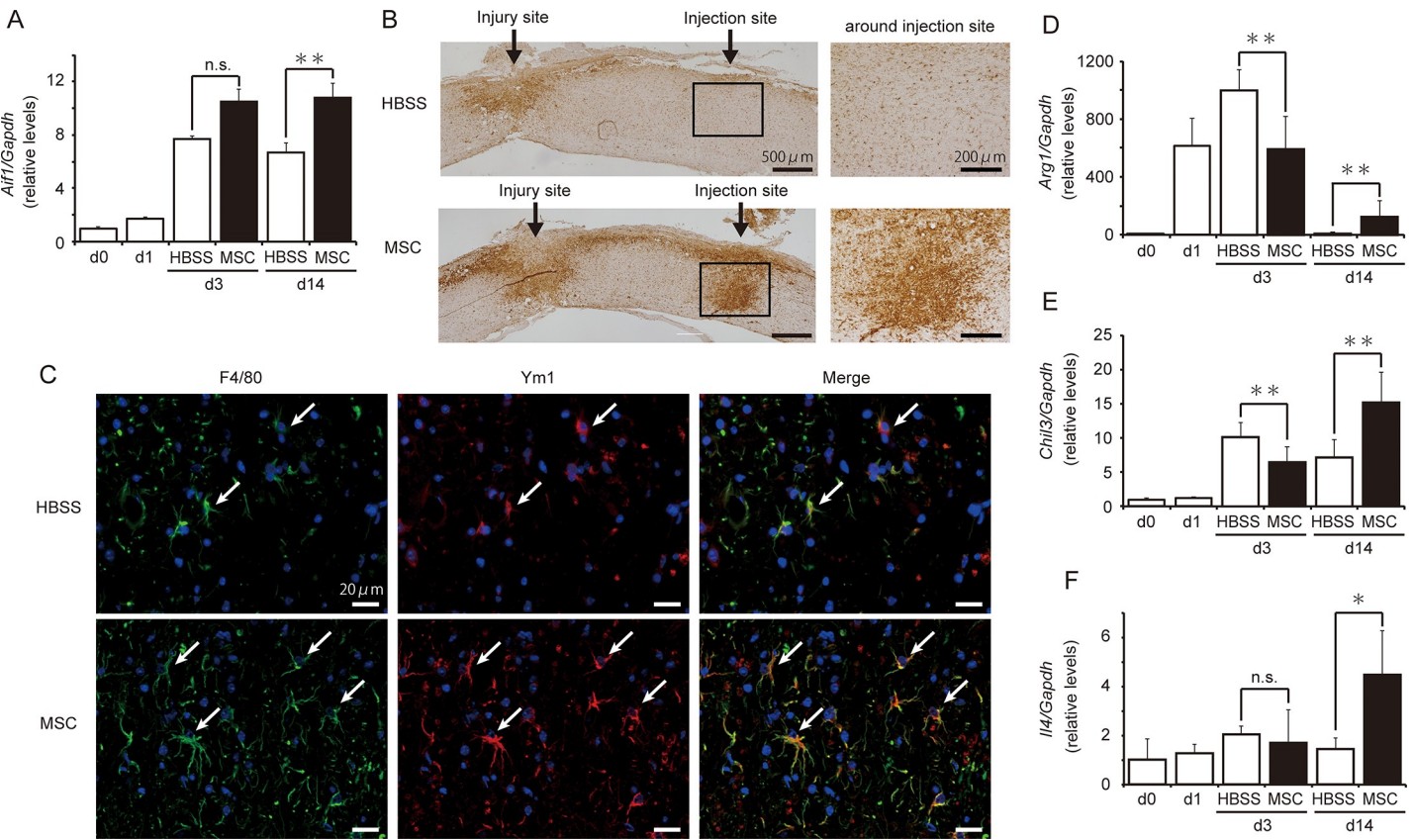

**Fig 7. Human mesenchymal stem/stromal cells (hMSCs) increased levels of alternatively activated macrophages (AAM) and anti-inflammatory cytokines. (A)** The transplantation of hMSCs significantly increased the levels of *Aif1* expression on postoperative day (pod) 14 (n = 9 mouse) relative to those observed in the HBSS-treated animals (n = 8 mouse). Data are expressed as the mean ± SEM. **: $P < 0.01$ (Dunnett's *post hoc* test), n.s.: not significant. **(B)** Iba1-positive MG/MΦ were clustered around the site of hMSC injection on pod 14 (n = 3 mouse in each group). **(C)** F4/80+ cells (*green*) exhibited greater overlap with Ym1+ cells (*red*) around the injection site in the hMSC-treated group than in the HBSS-treated group. Nuclear staining was performed using DAPI. Transplantation of hMSCs significantly increased the expression of the AAM markers *Arg1* **(D)** and *Chil3* **(E)**, as well as that of the anti-inflammatory cytokine *IL-4* **(F)** on pod 14. All data are expressed the relative level (fold) to compare with pod 0 (basal level, n = 6 mouse) after normalization against *Gapdh* as a housekeeping gene. Data are expressed as the mean ± SEM. *: $P<0.05$, **: $P < 0.01$ (Dunnett's *post hoc* test), n.s.: not significant.

during the experimental period. However, *Dpysl2* expression increased on pod 14 and was significantly higher in hMSC-treated mice than in HBSS-treated mice ([Fig 8A]). Similar tendencies were observed for *Gap43* expression, which was significantly higher in hMSCs-treated animals on pod 14 ([Fig 8B]). CRMP2+ and GAP43+ immunoreactions were detected in the peri-injury region and clearly merged with NeuN+ neurons on pod 14 ([Fig 8C and 8D]).

## CCL5 promotes AAM activity and axonal regeneration

To confirm whether increases in *Ccl5* expression following hMSC injection contribute to MG/MΦ polarization and axonal regeneration, mouse recombinant CCL5 was injected into the spinal cord after SCI on pod 7. On the following day, we examined the gene expression of AAM and axonal regeneration markers. *Aif1* expression remained unchanged following CCL5 injection ([Fig 9A]). However, CCL5-treated mice tended to increase the expression of *Chil3* and *Arg1* and exhibited significantly greater *Chil3* expression than did vehicle-treated mice ($p = 0.004$) ([Fig 9B and 9C]), suggesting that CCL5 contributes to MG/MΦ polarization.

CCL5-injected mice also exhibited increases in *Dpysl2* ($p = 0.097$) and *Gap43* ($p = 0.033$) expression ([Fig 9D and 9E]), suggesting that CCL5 also contributes to axonal regeneration.

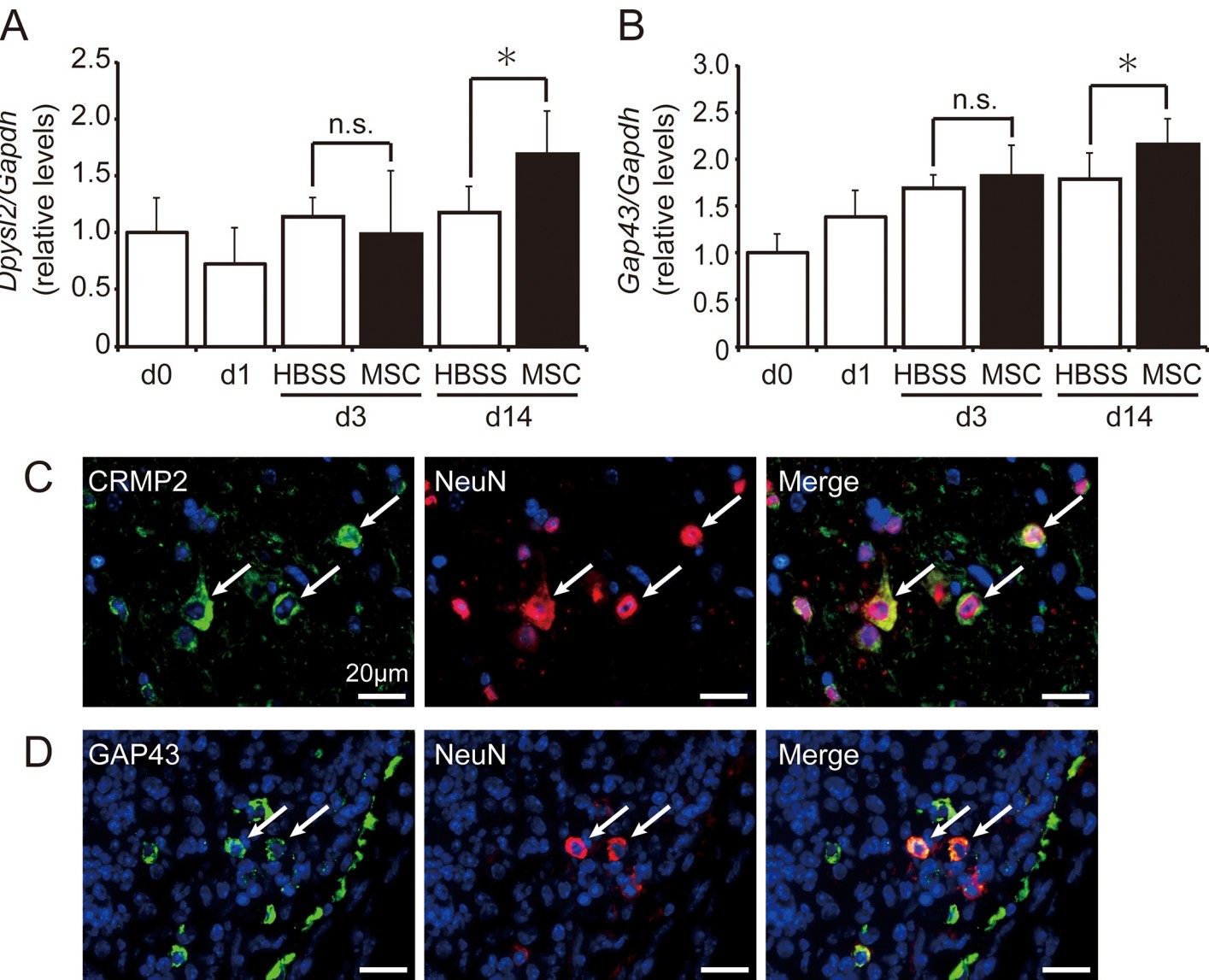

**Fig 8. Human mesenchymal stem/stromal cells (hMSCs) promoted axonal regeneration.** The transplantation of hMSCs increased the expression of the axonal regeneration markers *Dpysl2* (**A**) and *Gap43* (**B**). Mice treated with hMSCs (n = 9 mouse) exhibited significantly greater gene expression than did the HBSS-treated mice (n = 8 mouse) on postoperative day (pod) 14. All data are expressed the relative level (fold) to compare with pod 0 (basal level, n = 6 mouse) after normalization against *Gapdh* as a housekeeping gene. Data are expressed as the mean ± SEM. *: *P < 0.05* (*Student's t*-test), n.s.: not significant. (**C**) CRMP2 immunoreactivity (*green*) was observed in NeuN+ neurons (*red*) at the peri-injury site on pod 14. (**D**) GAP43 immunoreactivity (*green*) was merged with NeuN+ neurons (*red*) at the peri-injury site on pod 14 (n = 3 mouse in each group). *Blue* indicates nuclear staining using DAPI.

## Discussion

Human-MSC therapy has shown promise in supporting regeneration and/or recovery from diverse tissue injuries, including neuronal injuries [3–6]. Although we have reported that implanted hMSCs suppress neural injuries in mice by modulating MG/MΦ after brain ischemia and SCI [7, 8], the mechanism and form of communication between hMSCs and MG/MΦ remain unclear. In the present study, we examined the influences of hMSCs on chemokine expressions because chemokines guide the migration of immune cells, including MΦ. Using our SCI model, we especially focused on hMSC-induced changes in two MΦ-related chemokine axes: CCL2/CCR2 and CCL5/CCR5. We here xenografted hMSCs into mouse spinal cord

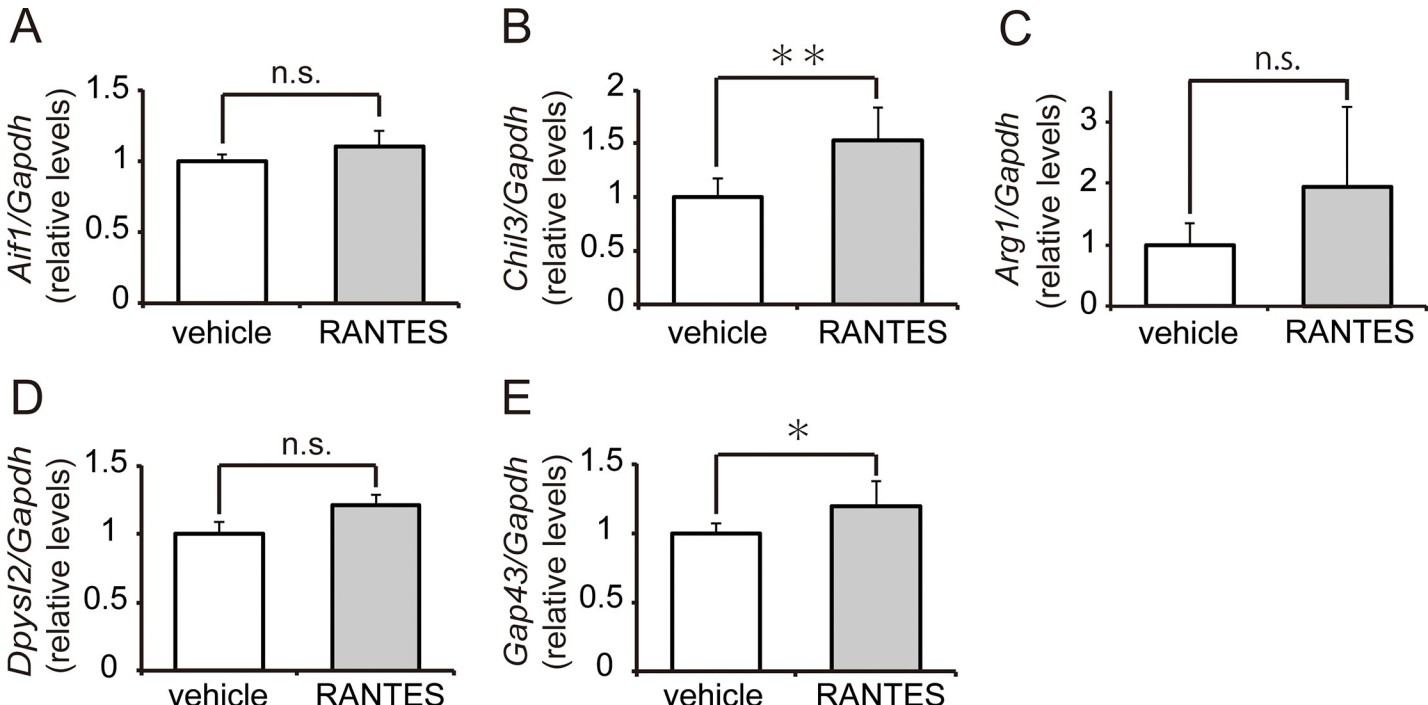

**Fig 9. Microglia/macrophage (MG/MΦ) polarization and axonal regeneration were enhanced by CCL5.** To determine whether increases in CCL5 following hMSC transplantation contribute to MG/MΦ polarization and axonal regeneration, recombinant mouse CCL5 (rmCCL5, n = 6 mouse) or vehicle (n = 6 mouse) was injected into the spinal cord on postoperative day (pod) 7. We examined levels of gene expression for MG/MΦ polarization (**A-C**) and axonal regeneration (**D and E**) 24 hours after injection. There were no obvious differences in the levels of the pan-macrophage marker *Aif1* (**A**). Expression of the M2-type AAM marker *Chil3* (**B**) significantly increased following the injection of rmCCL5 (*p<0.01*). Although *Arg1* expression (**C**) also tended to increase, these increases were nonsignificant. Expression of the axonal regeneration marker *Gap43* (**E**) significantly increased following the injection of rmCCL5 (*p<0.05*), while *Dpysl2* (**D**) expression tended to increase (*p = 0.097*). All data are expressed the relative level (fold) to compare with vehicle treated animals (n = 6 mouse) after normalization against *Gapdh* as a housekeeping gene. Data are expressed as the mean ± SEM. **: *P < 0.01*, *: P < 0.05 (Dunnett's *post hoc* test).

without administration of immunosuppressants to not influence an immune response. Many studies were given immunosuppressants for decrease of the rejection by xenografting. However, we have previously reported that injection of the hMSCs into immunocompetent mouse brain did not change the fate of cells to compare with immunodeficient mice [8]. These are supported and reproduced by many studies [10].

The *Ccl2* and *Ccr2* expressions drastically increased on pod1, and CCL2[+] immunoreactions were recognized in neurons in the acute phase following SCI. While previous studies have reported that CCR2 is mainly expressed in monocytes and infiltrated MΦ [34], CCR2[+] reactions had barely merged with F4/80[+] MG/MΦ; however, CCR2[+] reactions overlapped greatly with Ly-6G[+] granulocytes, which were likely neutrophils, at the epicenter of the injury on pod 1. Aberrations in the CCL2/CCR2 axis are reportedly associated with neuroinflammatory disease. [35–39]. CCR2 gene-deficient (KO) animals exhibit decreases in lesion size and recruitment of monocytes into the lesion following traumatic brain injury (TBI) [18] and SCI, [40]. Moreover, CCR2[+] neutrophils are activated under inflammatory conditions, thereby exacerbating the inflammatory response [41, 42]. According to these studies, most CCR2[+]/ Ly-6G[+] granulocytes on pod 1 exhibited co-localization of iNOS, suggesting an increase of inflammation.

In the present study, the hMSC implantation resulted in improvements in locomotor activity and tended to decrease lesion size; these findings were consistent with our previous report [7]. While we initially expected that the implanted hMSCs would diminish *Ccl2* expression,

the levels of *Ccl2*, and not those of *Ccr2*, increased following hMSC transplantation. However, the numbers of Ly-6G[+] and CCR2[+]/Ly-6G[+] cells did not change significantly with hMSC transplantation. Interestingly, CCR2[+] immunoreactions after hMSC transplantation were observed in NeuN[+]-neurons as well as Ly-6G[+] cells.

Some recent studies have speculated that, in addition to its role in chemotaxis, CCL2 exerts neuroprotective and immunomodulatory effects [18, 19]. CCL2 aids in the migration of neural precursor cells and promotes neuroregeneration in patients with TBI [39] and stroke [43]. Kwon *et al.* suggested that the overexpression of CCL2 derived from neurons promotes axonal regeneration via AAM activation after CNS injury [44]. Investigating the condition of inflammation and neutrophil activation, we found that the transplantation of hMSCs decreased the gene expression levels of *Il1b*, *Elane* and *Mpo*; these observations indicated reduced inflammation and neutrophil activation. Recent findings are consistent with our results: neutrophils polarized after infiltration can exert both beneficial and deleterious effects following tumorigenesis [45–47], stroke [48] and TBI [49]. Further, we found that mice treated with hMSCs exhibited persistent *Ccl2* expression for 2 weeks as well as increases the expression of *Zc3h12a*, which encodes MCPIP1 and is known to induce by CCL2 and IL-1β, on pod14. MCPIP1 plays a role in anti-inflammation via the down-regulation of NF-κB signaling [50], and MCPIP1-KO mice reportedly exhibit significant increases in infarct volume after brain ischemia [51]. Although further studies are required to determine the precise roles of chronic increases in CCL2, the enhancement of *Ccl2* expression via hMSC transplantation may contribute to anti-inflammation.

We also examined the expression of *Ccl5* after inducing SCI: it drastically increased by pod 14. While CCL5 binds to three receptors, CCR1, CCR3, and CCR5, we focused on CCR5 because *Ccr5* expression remained significantly elevated through pod 14. Our findings vary in their consistency with previous research, as the role of the CCL5/CCR5 axis in CNS injuries is controversial in the literature. CCL5 deletion reduces infarct volume in cerebral ischemia [52], and CCR5 blockade improves locomotor function in a murine model of SCI [53]. These observations suggest that CCL5 acts as a deleterious chemokine. Conversely, CCR5-KO mice exhibit increases in infarct volume in models of ischemia [54] and accelerated motor neuronal death in models of hypoglossal nerve injury [55]. Application of CCL5 to cultured neural cells significantly suppressed cell death and increased the expression of some growth factors [21]. Moreover, CCL5 decreased IL-1β, IL-6, tumor necrosis factor alpha (TNF-α), and iNOS gene expression in lipopolysaccharide-treated primary MG cultures [55].

The present study found that the hMSC implantation significantly increased *Ccl5* expression by pod 14 but observed no changes in *Ccr5* expression. Previous investigations have reported that CCL5 expression in neurons, MΦ, and astrocytes of the spinal cord of models of neuropathic pain and ischemia increased from pod 3 [21, 56]. We observed that CCL5 and CCR5 localized to neurons and MG/MΦ; this was not affected by hMSC implantation. However, the intervention increased the expressions of pan-MG/MΦ (*Aif1*) and M2-type AAM (*Arg1* and *Chil3*) markers. Moreover, the animals that received the hMSC implantation exhibited increases in *Il4* expression by pod 14. IL-4 is a major activator of AAM polarization [57]. Interestingly, CCL2, CCL5, and insulin-like growth factor 1 (IGF-1) are also reportedly involved in the differentiation of MG/MΦ toward AAMs in osteoblasts [24], suggesting that greater hMSC-induced *Il4* and *Ccl5* expression by might stimulate MG/MΦ migration and polarization.

Axonal extension and reconstruction of neuronal networks are important events in the recovery of spinal function. Research has suggested that AAMs promote axonal growth and overcome the effects of inhibitory substrates [58]. Several studies have also indicated that the implantation of AAMs into the injured spinal cord induces axonal regrowth and functional

improvement [59, 60]. In the present study, implantation of hMSCs increased the expression of the genes encoding axonal regenerative markers, including *Dpysl2* and *Gap43*, on pod 14, and corresponding immunoreactions were observed in the neurons at the peri-injury site. To obtain direct evidence for the contribution of the increased *Ccl5* expression to axonal extension, we injected rmCCL5 into the spinal cord on pod 7 and determined the gene expression of M2-type AAM as well as the levels of axonal regenerative markers. We observed significantly greater *Chil3* and *Gap43* expression in rmCCL5-treated mice, which also tended to exhibit increases in *Arg1* and *Dpysl2* expression; however, no obvious changes in *Aif1* expression were observed following CCL5 injection. These results suggest that the CCL5 may partially participate in AAM polarization and axonal extension.

In the present study, we examined the role of hMSCs after SCI by focusing on two representative MΦ-related chemokine axes, CCL2/CCR2 and CCL5/CCR5, both of which exhibited acute or chronic increases following SCI. hMSC implantation upregulated the levels of these chemokines. We also confirmed improvements in the clinical symptoms of SCI following hMSC injection. While the CCL2/CCR2 axis contributes to the enhancement of inflammation, hMSC-mediated increases in CCL2 did not alter the number of granulocytes. Moreover, hMSCs might induce MG/MΦ polarization to the M2-type AAM and drive axonal generation. While chemokines could participate in these functions, the direct mechanisms by which the communication between hMSCs and the recipient tissues yield these phenomena remain to be elucidated. Future studies should examine mice subjected to chemokine KO or gene silencing to determine whether hMSCs influence CCL2/CCR2 and CCL5/CCR5 axes in post-SCI neuroregeneration.

## Conclusions

In conclusion, the post-SCI implantation of hMSCs enhances the expression of *Ccl2* and *Ccl5* and improves locomotor activity. Moreover, the implantation of hMSCs enhances MG/MΦ polarization to AAM and the gene expression of axonal regenerative markers. The function of hMSCs might be partially mediated by the chemokine axes.

## Supporting information

**S1 Checklist.**
(DOCX)

## Acknowledgments

We would like to thank Editage (www.editage.jp) for English language editing.

## Author Contributions

**Conceptualization:** Kazumichi Yagura, Tomomi Tsumuraya, Atsushi Sato.

**Data curation:** Kazumichi Yagura, Hirokazu Ohtaki, Naoto Kawada, Keisuke Suzuki, Motoyasu Nakamura.

**Formal analysis:** Kazumichi Yagura, Hirokazu Ohtaki, Kazuyuki Miyamoto.

**Funding acquisition:** Hirokazu Ohtaki, Atsushi Sato, Yutaka Hiraizumi.

**Investigation:** Kazumichi Yagura, Hirokazu Ohtaki, Tomomi Tsumuraya, Atsushi Sato.

**Methodology:** Kazumichi Yagura, Hirokazu Ohtaki.

**Project administration:** Kazumichi Yagura, Hirokazu Ohtaki, Masahiko Izumizaki, Yutaka Hiraizumi, Kazuho Honda.

**Resources:** Hirokazu Ohtaki, Atsushi Sato, Yutaka Hiraizumi.

**Software:** Kazumichi Yagura, Hirokazu Ohtaki.

**Supervision:** Hirokazu Ohtaki, Koji Kanzaki, Kenji Dohi, Masahiko Izumizaki, Yutaka Hiraizumi, Kazuho Honda.

**Validation:** Kazumichi Yagura, Hirokazu Ohtaki, Tomomi Tsumuraya, Atsushi Sato, Kazuyuki Miyamoto, Naoto Kawada, Keisuke Suzuki, Motoyasu Nakamura, Koji Kanzaki, Masahiko Izumizaki, Yutaka Hiraizumi, Kazuho Honda.

**Visualization:** Kazumichi Yagura, Hirokazu Ohtaki.

**Writing – original draft:** Kazumichi Yagura, Hirokazu Ohtaki.

**Writing – review & editing:** Kazumichi Yagura, Hirokazu Ohtaki, Koji Kanzaki, Masahiko Izumizaki, Yutaka Hiraizumi, Kazuho Honda.

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
