## [Decision Letter · Decision Letter 0]

13 Dec 2019

PONE-D-19-28233

The enhancement of CCL2 and CCL5 by human bone marrow-derived mesenchymal stem/stromal cells might contribute to inflammatory suppression and axonal extension after spinal cord injury.

PLOS ONE

Dear Dr Ohtaki,

Thank you for submitting your manuscript to PLOS ONE. After careful consideration, we feel that it has merit but does not fully meet PLOS ONE’s publication criteria as it currently stands. Therefore, we invite you to submit a revised version of the manuscript if you think you can satisfactorily addresses the points raised during the review process.

In addition to the concerns from reviewer 1 (see details below), please address following:

1. Provide evidence that hMSCs are present at the injection site

2. What is the survival rate and where are they located if they are not at the injury site

3. Why are there no dendrites on neurons?

We would appreciate receiving your revised manuscript by Jan 27 2020 11:59PM. To enhance the reproducibility of your results, we recommend that if applicable you deposit your laboratory protocols in protocols.io, where a protocol can be assigned its own identifier (DOI) such that it can be cited independently in the future. For instructions see: http://journals.plos.org/plosone/s/submission-guidelines#loc-laboratory-protocols

We look forward to receiving your revised manuscript.

Kind regards,

Xing-Ming Shi, Ph.D

Academic Editor

PLOS ONE

Journal Requirements:

3. Your ethics statement must appear in the Methods section of your manuscript. If your ethics statement is written in any section besides the Methods, please move it to the Methods section and delete it from any other section. Please also ensure that your ethics statement is included in your manuscript, as the ethics section of your online submission will not be published alongside your manuscript.

Reviewers' comments:

Reviewer's Responses to Questions

**Comments to the Author**

1. Is the manuscript technically sound, and do the data support the conclusions?

Reviewer #1: No

2. Has the statistical analysis been performed appropriately and rigorously? 

Reviewer #1: No

3. Have the authors made all data underlying the findings in their manuscript fully available?

Reviewer #1: Yes

4. Is the manuscript presented in an intelligible fashion and written in standard English?

Reviewer #1: No

5. Review Comments to the Author

Reviewer #1: Human bone marrow-derived mesenchymal stem/stromal cell (hMSC) transplantation is a promising strategy for improving functional recovery following traumatic spinal cord injury (SCI). However, the mechanism behind the integration of hMSCs and their beneficial effects remains elusive. The authors seek to 1) transcriptionally characterize the role of microglia/macrophages after SCI; 2) identify the cellular localization of CCL2/CCR2 and CCL5/CCR5 chemokine systems; 3) assess the role of hMSCs on CCL2/CCR2 and CCL5/CCR5; and 4) validate the role of CCL5 administration on microglia/macrophages polarization. The authors suggest that hMSCs increase CCL2 and CCL5 expression, increase gene expression of axonal regenerative markers, and increase microglia/macrophage polarization toward the alternatively activated phenotype.

Major Comments:

• The main objective of the study needs to be clearly indicated and experiments need to coherently follow the main objective. How were the genes selected for the PCR experiment? Why choose CCL2/CCR2 and CCL5/CCR5?

• The representation of expressional changes in Figure 1 needs to be revised. Line graphs or heatmaps would be more suitable for representing time-course transcriptional differences. How was the absolute PCR analysis performed?

• The functional recovery assessment is singular (relies only on BMS scores) and is terminated at 14 days while the difference between the treatment is continuing to deviate. The authors need to provide a justification for not continuing their analyses to the chronic phase.

• The authors need to indicate how their transcriptional work in Figure 1 advances the previous studies performing transcriptional analyses in SCI research.

• It’s not clear to the reader why the authors refrained from administration of immunosuppressants or immunodeficient animals, particularly considering the low survival of hMSCs following transplantation.

Minor Comments:

• The authors need to justify their choice for using a transection model for studying immune cell migration.

• The abstract needs to be reorganized and compressed. While the authors are informative and descriptive, they should succinctly describe their major findings and main inferences.

• Line 25: “Central” should not be capitalized.

6. PLOS authors have the option to publish the peer review history of their article (what does this mean?). If published, this will include your full peer review and any attached files.

Reviewer #1: No

---

## [Author Response · Author response to Decision Letter 0]

2 Jan 2020

Thank you very much for the academic editor’s and reviewer’s comments to improve our manuscript of paper (PONE-D-19-28233). We read the comments well, and answered and revised according to your suggestions. We also removed all the term “data not shown” as journal requirements.

Comments from editor

1. Provide evidence that hMSCs are present at the injection site

2. What is the survival rate and where are they located if they are not at the injury site

Thank you very much for your comments. We thought that the comments 1 and 2 linked each other. So, we answered them together. 

Injected hMSCs were disappeared quickly from the injective site. However, it is known to rescue several diseases (Prockop DJ. Clin Pharmacol Ther. 2007; https://ascpt.onlinelibrary.wiley.com/doi/full/10.1038/sj.clpt.6100313). 

Also. in present study, we determined the fate of hMSCs using human GAPDH mRNA (hGAPDH) expression in the mouse spinal cord by real-time PCR (Fig. 1A). The estimate number of hMSCs was calculated by standard curve of the gene expression. Standard curves were generated by adding serial dilutions of hMSCs (1 × 102 to 1 × 106 cells) to spinal cord samples (T7 to T12 vertebrae) of uninjured mice prior to homogenization (see page 13, line 95~). 

We have also reported previously the retention of hMSCs by hAlu genomic DNA expression and the localization of PKH26-labeled hMSCs by immunohistochemistry in same animal model (Fig. 2 in Tsumuraya et al. J Neuroinflammation 2015; https://jneuroinflammation.biomedcentral.com/articles/10.1186/s12974-015-0252-5). We showed in the study that the injected hMSCs are seen to have migrated along the spinal cord toward the injury site at 7 days after SCI. 

The fate of hMSCs in the present study was shown in Fig. 1A. Immediately after hMSC injection (pod 1), approximately 260,000 hMSCs were detected (representing 50% of the injected cells) including the site of injection. The numbers of hMSCs decreased to approximately 40,000 cells on pod 3(15% of pod1), and have few-to-no hMSCs were detected on pod 14 (see page 21, line 10~). The reason, we choose hGAPDH, but not hAlu in present study is to eliminate a possibility of genomic DNA from hMSCs debris although the result was almost similar with previously. 

We have reported the fate of hMSCs in our other studies of brain ischemia (Ohtaki et al. Proc Natl Acad Sci U S A. 2008; https://www.pnas.org/content/105/38/14638.long) and STZ-induced diabetes (Murai et al. PLoS One 2017; https://journals.plos.org/plosone/article?id=10.1371/journal.pone.0186637). These studies also support in the present results. 

The question of “where are they located if they are not at the injury site” is very difficult to answer. In the present study, the cells as the hGAPDH level almost disappeared from the spinal cord on pod 14. The cells might eliminate by phagocytes and/or move to somewhere. We have examined the distribution of hMSCs in systemic organs by an in vivo fluorescence imager (Murai et al. PLoS One 2017). However, the detection of small amount of cells was extremely difficult. So, I am sorry we do not have accurate answer of this. 

3. Why are there no dendrites on neurons?

In present study, we used anti-Neu N antibody as a neuronal marker. Neu N is known to be stained neuronal nuclei and the peri-nucleic region (nerve cell body), but not or few in fibers. So the staining of neurons seems to round and triangle sometimes. 

Comments to reviewer

Thank you very much for your supportive comments. We revised our manuscript (indicated yellow highlight in the text) and commented to you below. 

Major Comments:

1. The main objective of the study needs to be clearly indicated and experiments need to coherently follow the main objective. How were the genes selected for the PCR experiment? Why choose CCL2/CCR2 and CCL5/CCR5?

In present study was designed according to our previous studies. We have focused on the communication between transplanted hMSCs and microglia/macrophages (MG/MΦ) on SCI. Therefore, we firstly selected mainly the MG/MΦ including monocyte-related chemokines and the receptors. Then, from expressing pattern we further focused on the CCL2/CCR2 and CCL5/CCR5. CCL5/CCR5 is a chemokine and receptors increased at chronic periods. Although many chemokines (receptors) increased at acute periods, we selected the most popular MG/MΦ-related chemokine CCL2/CCR2 axis. We think that we should examine the other chemokines in the future study.

Although this is unpublished data, we are recognizing in another project that CCL2/CCR2 axis is an important signal to communicate hMSCs and recipient cells. Would you wait for wait a little more to reveal our next publication? 

2. The representation of expressional changes in Figure 1 needs to be revised. Line graphs or heatmaps would be more suitable for representing time-course transcriptional differences. How was the absolute PCR analysis performed?

Thank you very much. We revised bar to line graphs in Fig.1 (now, this is Fig.2). Moreover, we have not described the unit in the text including the legend and graphs clearly. They are relative level (fold) to compare with day 0. This mentioned in the methods section (see page 14 line 125~) and figure legend. As described below, we reconsidered the order of the figures from your comment #3. Fig.1 in original manuscript changed to Fig.2.

3. The functional recovery assessment is singular (relies only on BMS scores) and is terminated at 14 days while the difference between the treatment is continuing to deviate. The authors need to provide a justification for not continuing their analyses to the chronic phase.

We have reported injection of hMSCs suppress neural injury through the macrophage or microglial modulation in the acute phase (within a week) after ischemia (Ohtaki et al. Proc Natl Acad Sci U S A. 2008). Moreover, the phenomenon has been reproduced after SCI as well (Tsumuraya et al. J Neuroinflammation 2015). Therefore, we decided the experiment periods based on the previous our studies and checked the reproducibility by BMS and gene expression of neuron specific enolase (Eno2) in the present study. However, we thought your comments also very reasonable if we showed figures in present order. We reconsidered again the order of the figures. We inserted Fig. 4 prior to Fig. 1 to be logically. Thank you for your supportive comments.

4. The authors need to indicate how their transcriptional work in Figure 1 advances the previous studies performing transcriptional analyses in SCI research.

Yes. Some of SCI studies have reported to the gene expression of chemokines. However, we have not determined the expressions in this model so far. Therefore, we did them.

5. It’s not clear to the reader why the authors refrained from administration of immunosuppressants or immunodeficient animals, particularly considering the low survival of hMSCs following transplantation.

Thank you for your comments. As your suggestion, many studies administer administration of immunosuppressants or use immunodeficient animals. However, to observe inflammation or immune systems, we were afraid that they might interfere to the systems. We have previously examined injection of hMSCs into brain parenchyma and compared the fate of hMSCs between immunodeficient and immunocompetent animals. Resulting, no significant differences were recognized (Ohtaki et al. Proc Natl Acad Sci U S A. 2008). Moreover, repetitive injections of hMSCs into immunocompetent animals did not show any anaphylactic reactions or die (Murai et al. PLoS One 2017). The following manuscript explain the reason to inject hMSCs into immunocompetent animals. We added this in our references.

Prockop DJ, Oh JY, Lee RH. Data against a Common Assumption: Xenogeneic Mouse Models Can Be Used to Assay Suppression of Immunity by Human MSCs. Mol Ther. 2017 2;25(8):1748-1756. doi: 10.1016/j.ymthe.2017.06.004.

Minor Comments:

6. The authors need to justify their choice for using a transection model for studying immune cell migration.

To compare with contusion model, the transection model is localized the injury to a small region. Therefore, we thought to be clear cell-cell communications. Moreover, as the other reason, we continuously studied SCI using this model and there are some evidences for the model.

7. The abstract needs to be reorganized and compressed. While the authors are informative and descriptive, they should succinctly describe their major findings and main inferences.

Thank you. We revised.

8. Line 25: “Central” should not be capitalized.

Thank you. We revised.

---

## [Decision Letter · Decision Letter 1]

28 Jan 2020

PONE-D-19-28233R1

The enhancement of CCL2 and CCL5 by human bone marrow-derived mesenchymal stem/stromal cells might contribute to inflammatory suppression and axonal extension after spinal cord injury.

PLOS ONE

Dear Dr Ohtaki,

Thank you for submitting your manuscript to PLOS ONE. After careful consideration, we feel that it has merit but does not fully meet PLOS ONE’s publication criteria as it currently stands. Therefore, we invite you to submit a revised version of the manuscript that addresses the points (details below).

We would appreciate receiving your revised manuscript by Mar 13 2020 11:59PM. To enhance the reproducibility of your results, we recommend that if applicable you deposit your laboratory protocols in protocols.io, where a protocol can be assigned its own identifier (DOI) such that it can be cited independently in the future. For instructions see: http://journals.plos.org/plosone/s/submission-guidelines#loc-laboratory-protocols

We look forward to receiving your revised manuscript.

Kind regards,

Xing-Ming Shi, Ph.D

Academic Editor

PLOS ONE

Reviewers' comments:

Reviewer's Responses to Questions

**Comments to the Author**

1. If the authors have adequately addressed your comments raised in a previous round of review and you feel that this manuscript is now acceptable for publication, you may indicate that here to bypass the “Comments to the Author” section, enter your conflict of interest statement in the “Confidential to Editor” section, and submit your "Accept" recommendation.

Reviewer #1: (No Response)

2. Is the manuscript technically sound, and do the data support the conclusions?

Reviewer #1: Yes

3. Has the statistical analysis been performed appropriately and rigorously? 

Reviewer #1: Yes

4. Have the authors made all data underlying the findings in their manuscript fully available?

Reviewer #1: Yes

5. Is the manuscript presented in an intelligible fashion and written in standard English?

Reviewer #1: Yes

6. Review Comments to the Author

Reviewer #1: The manuscript is much improved. However, grammar mistakes and the writing style at the end of the introduction should be addressed. Additionally, the authors may want to mention the justification for the transection model and not administering immunosuppressants in the discussion.

7. PLOS authors have the option to publish the peer review history of their article (what does this mean?). If published, this will include your full peer review and any attached files.

Reviewer #1: No

---

## [Author Response · Author response to Decision Letter 1]

7 Feb 2020

Thank you very much for the academic editor’s and reviewer’s comments to improve our manuscript of paper (PONE-D-19-28233). We read the comments well, and revised according to your suggestions.

Comments to reviewer

1. The manuscript is much improved. However, grammar mistakes and the writing style at the end of the introduction should be addressed. Additionally, the authors may want to mention the justification for the transection model and not administering immunosuppressants in the discussion.

Thank you very much for your supportive comments. We revised our manuscript (indicated yellow highlight in the text) according to your comments.

Moreover, we checked entire of manuscript again and slightly changed the order of references.

---

## [Editor Report · Decision Letter 2]

21 Feb 2020

The enhancement of CCL2 and CCL5 by human bone marrow-derived mesenchymal stem/stromal cells might contribute to inflammatory suppression and axonal extension after spinal cord injury.

PONE-D-19-28233R2

Dear Dr. Ohtaki,

We are pleased to inform you that your manuscript has been judged scientifically suitable for publication and will be formally accepted for publication once it complies with all outstanding technical requirements.

With kind regards,

Xing-Ming Shi, Ph.D

Academic Editor

PLOS ONE
---

## [Editor Report · Acceptance letter]

25 Feb 2020

PONE-D-19-28233R2 

The enhancement of CCL2 and CCL5 by human bone marrow-derived mesenchymal stem/stromal cells might contribute to inflammatory suppression and axonal extension after spinal cord injury. 

Dear Dr. Ohtaki:

I am pleased to inform you that your manuscript has been deemed suitable for publication in PLOS ONE. Congratulations! Your manuscript is now with our production department. 

With kind regards,

on behalf of

Dr Xing-Ming Shi 

Academic Editor

PLOS ONE